# Glyphosate inhibits melanization and increases susceptibility to infection in insects

**Daniel F. Q. Smith**[1], **Emma Camacho**[1], **Raviraj Thakur**[2☉¤], **Alexander J. Barron**[3☉], **Yuemei Dong**[1], **George Dimopoulos**[1], **Nichole A. Broderick**[3‡], **Arturo Casadevall**[1‡¤*]

**1** W. Harry Feinstone Department of Molecular Microbiology and Immunology, Johns Hopkins Bloomberg School of Public Health, Baltimore, Maryland, United States of America, **2** Department of Otolaryngology, Head and Neck Surgery, Johns Hopkins Medicine, Baltimore, Maryland, United States of America, **3** Department of Biology, Johns Hopkins University, Baltimore Maryland, United States of America

☉ These authors contributed equally to this work.
¤ Current address: Cancer Early Detection Advanced Research Center, Knight Cancer Institute, Oregon Health and Science University, Portland, Oregon, United States of America
‡ NAB and AC are joint senior authors on this work.
* acasade1@jhu.edu

**Data Availability Statement:** The datasets presented in this manuscript are publicly available Figshare data repository at: https://figshare.com/projects/Glyphosate_Inhibits_Melanization_and_

## Abstract

Melanin, a black-brown pigment found throughout all kingdoms of life, has diverse biological functions including UV protection, thermoregulation, oxidant scavenging, arthropod immunity, and microbial virulence. Given melanin's broad roles in the biosphere, particularly in insect immune defenses, it is important to understand how exposure to ubiquitous environmental contaminants affects melanization. Glyphosate—the most widely used herbicide globally—inhibits melanin production, which could have wide-ranging implications in the health of many organisms, including insects. Here, we demonstrate that glyphosate has deleterious effects on insect health in 2 evolutionary distant species, *Galleria mellonella* (Lepidoptera: Pyralidae) and *Anopheles gambiae* (Diptera: Culicidae), suggesting a broad effect in insects. Glyphosate reduced survival of *G. mellonella* caterpillars following infection with the fungus *Cryptococcus neoformans* and decreased the size of melanized nodules formed in hemolymph, which normally help eliminate infection. Glyphosate also increased the burden of the malaria-causing parasite *Plasmodium falciparum* in *A. gambiae* mosquitoes, altered uninfected mosquito survival, and perturbed the microbial composition of adult mosquito midguts. Our results show that glyphosate's mechanism of melanin inhibition involves antioxidant synergy and disruption of the reaction oxidation–reduction balance. Overall, these findings suggest that glyphosate's environmental accumulation could render insects more susceptible to microbial pathogens due to melanin inhibition, immune impairment, and perturbations in microbiota composition, potentially contributing to declines in insect populations.

## Introduction

Melanin, a black-brown pigment found in all biological kingdoms, is produced through a series of oxidation and reduction reactions. These reactions are typically catalyzed by 2 classes

Increases_Susceptibility_to_Infection_in_Insects/
99341. Additionally, the Figshare Digital Object
Identifiers (DOIs) for the individual figure datasets
are listed in S1 Table.

**Funding:** D.F.Q.S., E.C., A.C. are funded by National
Institute of Allergy and Infection Disease R01
AI052733 and Johns Hopkins Malaria Research
Institute (https://malaria.jhsph.edu/) Pilot Grant #
Casadevall_123. D.F.Q.S. is funded by National
Institutes of Health 5T32GM008752-18 and
1T32AI138953-01A1. A.J.B. and N.A.B. are funded
by National Institutes of Health R35GM128871.
The funders had no role in study design, data
collection and analysis, decision to publish, or
preparation of the manuscript. The salaries of D.F.
Q.S., A.C., and E.C. are in part funded by the
National Institute of Allergy and Infection Disease.
The salaries of D.F.Q.S., A.J.B., and N.A.B., are in
part funded by the National Institutes of Health. The
salary of E.C. is in part funded by Johns Hopkins
Malaria Research Institute.

**Competing interests:** The authors have declared
that no competing interests exist.

**Abbreviations:** ABTS, 2,2′-azino-bis(3-
ethylbenzothiazoline-6-sulfonic acid); AMPA,
aminomethylphosphonic acid; CFU, colony forming
unit; DPI, days postinfection; L-DOPA, 3,4-
dihydroxyphenylalanine; OTU, operational
taxonomic unit; PCoA, principal coordinates
analysis; ROS, reactive oxygen species.

of enzymes: laccases (EC. 1.10.3.2) and phenoloxidases—a family which includes tyrosinases (EC. 1.14.18.1) [1]. Tyrosinases are copper metalloenzymes found in bacteria, fungi, protists, arthropods, birds, and mammals [2–7], and have 2 catalytic roles: (1) hydroxylation of monophenols into *ortho*-diphenols, followed by (2) oxidation of *o*-catechols into *o*-quinones [8]. During melanization, tyrosinase converts 3,4-dihydroxyphenylalanine (L-DOPA) to dopaquinone. Dopaquinone undergoes oxidation and reduction reactions to first form dopachrome, then dihydroxyindole. Dihydroxyindole undergoes radical-mediated polymerization to form melanins [8,9].

Melanization is an important component of immunity in virtually all insects [9]. Upon infection, protease cascades are activated that cleave pro-phenoloxidases into active phenoloxidases. Phenoloxidases convert catecholamines in the hemolymph into melanin, which surrounds and eliminates the pathogen through exposure to reactive oxygen species (ROS) and lysis from toxic melanin intermediates [3,9–13]. This melanization process is a key component of insect immune defense against bacterial, fungal, and protozoan pathogens, nematode parasites, and insect parasitoids [14–21]. In addition, pathogens are cleared by 2 similar processes: nodulation of smaller microbes such as bacteria, fungi, and protozoa, and encapsulation for infections with larger organisms such as helminths and parasitoid eggs [22]. Both nodulation and encapsulation involve pathogen neutralization via melanin accumulation and hemocyte (insect "blood cells") aggregation around the pathogen [22]. Melanization and phenoloxidases are also important for wound healing and cuticular development—processes vital for insect health and survival [23,24]. Since melanization is an essential physiological process and effector of insect health, understanding how common environmental contaminants affect melanin production is important. The significance of this is also highlighted by findings suggesting that insect populations may be in decline in recent decades [25].

One ubiquitous chemical found in the environment is glyphosate, the most commonly used herbicide worldwide, which was previously shown to interfere with melanization in the fungus *Cryptococcus neoformans* [26]. Glyphosate is a phosphonic glycine analogue and the active ingredient in Roundup herbicides [27]. It kills plants through competitive inhibition of EPSP synthase in the shikimate pathway responsible for aromatic amino acid synthesis in many plants, fungi, and bacteria [28]. The low cost of glyphosate and wide availability of genetically modified glyphosate-resistant crops has increased both crop yields and glyphosate-based herbicide use in agriculture [29,30]. From 1996 to 2014, glyphosate-resistant crops were linked to a 12-fold global increase in glyphosate use, including 8-fold in the United States, 134-fold in Brazil, and 107-fold in Argentina [31,32].

In practice, glyphosate is commonly applied at concentrations of approximately 28 to 57 mM [33] or in formulations of 360 g/L (2 M), with 720 g (4 mol) per hectare [34]. Glyphosate-based herbicides are sprayed onto crops where the glyphosate is taken up by plant leaves and translocated to growing tissues throughout the plant [35]. Glyphosate is translocated to the roots where it is released into the soil [34]. In total, about 88% of the sprayed glyphosate ends up in the topsoil [36–38]. Less than 1% of glyphosate has been shown to enter water bodies, typically following heavy rain, snowmelt, ploughing, or erosion [37], but concentrations from <1 nM to approximately 30 μM in nearby water have been reported [39]. Further, glyphosate has been shown to enter the air through wind erosion and deposit via rain [40].

Glyphosate is remarkably stable, with half-life ranging from weeks to years depending on the surrounding microbial populations, which provide the primary mechanism of glyphosate degradation, while temperature, light, acidity, and salinity also play roles in the degradation process. Microbes mostly break down glyphosate into aminomethylphosphonic acid (AMPA), which persists up to 20 times longer than glyphosate and is often found in higher concentrations in topsoil and water [41–45].

While glyphosate may have harmful effects on microbes and animals (as reviewed in [35,46]), its impact on environmental microbial communities is inconclusive. Some studies demonstrate clear perturbation of microbial communities, including disrupting rhizosphere composition and fungal endophyte growth and viability [47–50], while others show little to no long-term impact on microbial communities [51–53], with no effects on overall soil health or reduction in soil microbial mass [54]. Microbial communities are also abundant in insect guts, where they are important for insect health [55–58], and several studies have linked detrimental effects of glyphosate on insect health to disruption of the microbiota. Honeybees exposed to glyphosate have altered microbiomes and are more susceptible to *Serratia marcescens* [59], although AMPA did not have the same effect [60]. In tsetse fly midguts, glyphosate disrupts *Wigglesworthia glossinidia's* production of folate—a compound required for tsetse fly health and vector competence for *Trypanosoma brucei* parasites [61].

Beyond effects on microbial communities, glyphosate has broad physiological impacts on insects, other arthropods, and vertebrates. While glyphosate was harmless to *Lepthyphantes tenuis* spiders, it changed behavior and increased mortality in *Pardosa milvina* and *Neoscona theisi* spiders [62–64]. Glyphosate reduced learning in *Aedes aegypti* mosquitoes [65] and in honeybees reduced survival and caused learning defects associated with feeding, homing, and flight behaviors [66,67]. Glyphosate and AMPA delayed development and reduced survival of the arthropod *Daphnia magna* [68]. Glyphosate induces oxidative stress and damage in many organisms, including *D. magna*, insects (fruit flies), amphibians (African clawed frog, European green toad, marsh frog), fish (brown trout, spotted snakehead fish), and mammals (rats) [69–74], often linked with lipid peroxidation and expression of antioxidant defenses (catalase, glutathione, and superoxide dismutase). In human erythrocytes, glyphosate and AMPA mixtures increase ROS production [75,76].

Given the melanin-inhibitory properties of glyphosate in fungi [26], we examined the roles of glyphosate and AMPA as inhibitors of insect melanization and phenoloxidase. We used 2 distinct insect models that both rely on melanin-based immunity: *Galleria mellonella*, a species of wax moths (Lepidoptera: Pyralidae), and *Anopheles gambiae*, a mosquito vector of malaria (Diptera: Culicidae). Considering melanin's importance in insect immunity, we evaluated glyphosate's effects on *G. mellonella* susceptibility to the pathogenic fungus *C. neoformans*, and on *A. gambiae* survival and susceptibility to the malaria parasite *Plasmodium falciparum* as measured by parasite oocyst burden. Additionally, we evaluated glyphosate's mechanism of melanin inhibition using L-DOPA auto-oxidation and mushroom tyrosinase-mediated oxidation models. Mushroom tyrosinase is commercially available and produces melanin in a similar mechanism as insect phenoloxidase. The purified enzyme and L-DOPA auto-oxidation allowed us to take a controlled step-by-step biochemical approach to show that glyphosate inhibits melanization by disrupting oxidative balance.

## Results

### Glyphosate and AMPA inhibit *Galleria mellonella* phenoloxidase activity

In insects, phenoloxidases are activated by serine proteases upon wounding or infection, thus triggering melanin production to either clot a wound or restrict a pathogen [10]. To investigate whether glyphosate inhibited insect melanogenesis, we used 2 models: *G. mellonella* wax moth larvae, and *A. gambiae* adult mosquitoes, a main vector of malaria.

In an ex vivo analysis using *G. mellonella* hemolymph, we found that glyphosate inhibited phenoloxidase activity in a dose-dependent manner, without addition of exogenous substrate (Fig 1A). Similar results were found with addition of a broad-spectrum protease inhibitor, which was used to control for continued activation of phenoloxidase, glyphosate-induced

cellular responses, and/or off-target effects on other components of the phenoloxidase cascade (S1A Fig). We also saw similar inhibition of phenoloxidase activity with the addition of exogenous L-DOPA; however, in these experiments, there was only a modest enhancement of phenoloxidase activity with lower glyphosate concentrations followed by striking inhibition at higher concentrations (S1B Fig). Importantly, glyphosate did not impact hemocyte viability, as measured by trypan blue exclusion (S1C Fig). AMPA, a primary breakdown product of glyphosate that accumulates in the environment [45], inhibited melanization similarly to glyphosate using *G. mellonella* hemolymph and a commercially available mushroom tyrosinase (Fig 1B, S1D Fig). These data show that glyphosate inhibits melanization in insects similar to what has been previously shown in fungi [26], thus indicating that glyphosate interferes with major melanin-based processes in at least 2 kingdoms of life.

## Glyphosate alters *G. mellonella* susceptibility to infection

Next, we sought to determine whether glyphosate increased in vivo susceptibility of *G. mellonella* larvae to foreign organisms. We injected *G. mellonella* final instar larvae with 2 μg of glyphosate (approximately 8 to 12 ng/mg per larvae) followed by infection with *C. neoformans* or a mock infection. The 2 mock infected groups, glyphosate-treated and phosphate-buffered saline (PBS)-treated, exhibited similar survival. However, in the infected groups, the glyphosate-treated larvae died faster compared to the PBS-treated controls (Gehan–Breslow–Wilcoxon test, $p = 0.013$) (Fig 1C). A similar, but nonsignificant, trend was seen with a *C. neoformans lac1Δ* strain (S1E Fig). The *lac1Δ* strain is unable to produce melanin, an important virulence factor in *C. neoformans* pathogenesis. This strain is less virulent in the *G. mellonella* model [78], potentially contributing to the lack of significant differences between the glyphosate and PBS-treated groups.

The decreased survival of the glyphosate-treated group infected with *C. neoformans* was correlated with smaller melanized particles within nodules formed during infection (in vivo), as compared to the PBS-treated group infected with *C. neoformans* (Fig 1D). For these experiments, we infected and drugged *G. mellonella* as we do during normal infections, then collected the hemolymph 24 hours later and imaged the nodules and aggregates that formed in vivo. The PBS-treated noninfected group had smaller or virtually no melanized structures. In 2 of 3 replicates, the PBS-treated infected group had more melanized structures than the glyphosate-treated group (S1G–S1I Fig). Further, in the glyphosate-treated infected group, we observed more *C. neoformans* cells, and those within nodules displayed lower degrees of melanin encapsulation compared to PBS-treated larvae (chi-squared test, $p = 0.0034$) (Fig 1E). The scoring was based on a system devised with 0 representing no melanin encapsulation of the yeast cell to 4 being the most melanin encapsulation, as depicted in Fig 1F. The one-time treatment with glyphosate did not disrupt time to larval pupation, a process mediated by laccases and phenoloxidases (S1F Fig). These data suggest a direct correlation between glyphosate treatment and increased susceptibility of *G. mellonella* to infection caused by decreased melanin-based immune response (nodulation).

## Glyphosate alters *A. gambiae* phenoloxidases, susceptibility to infection with malaria parasites, and survival

To ascertain the impact of glyphosate on *A. gambiae* mosquito melanization, we measured the phenoloxidase activity in whole-body mosquito homogenate following the addition of glyphosate. Similar to our results with *G. mellonella*, glyphosate inhibited phenoloxidase activity of *A. gambiae* homogenate in a dose-dependent manner (Fig 2A).

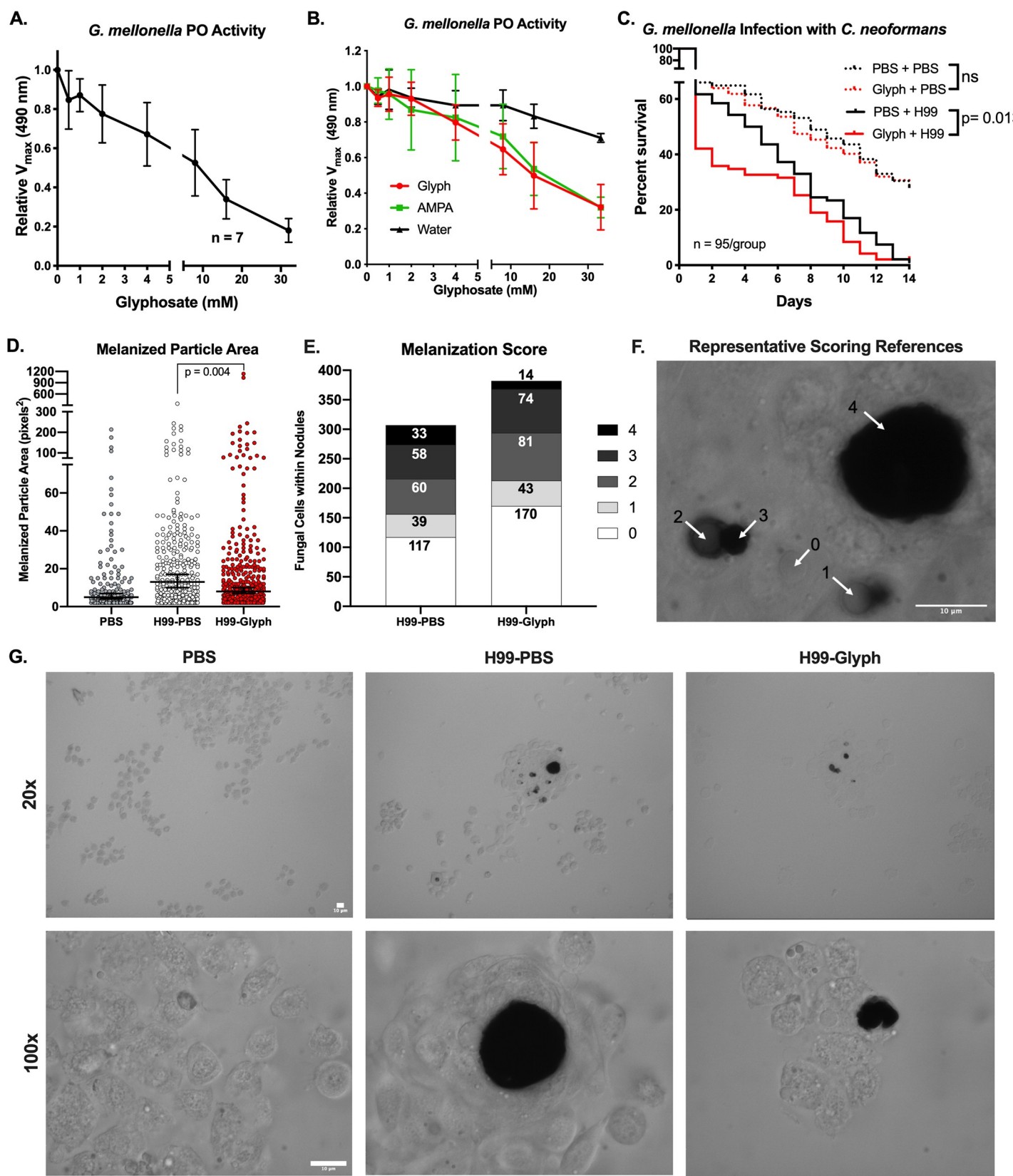

**Fig 1. Glyphosate inhibits *G. mellonella* melanization and increases infection susceptibility.** (**A**) Glyphosate inhibits the phenoloxidase activity of 1:10 dilutions of hemolymph without exogenously added L-DOPA. (**B**) AMPA, a primary metabolite of glyphosate, inhibits *G. mellonella* phenoloxidase-mediated melanization similar to glyphosate. Error bars in (**A, B**) represent ± SD (**C**) *G. mellonella* larvae drugged with glyphosate solution (10 μl of 1 mM) in PBS and infected 5 hours posttreatment with $10^4$ cells of WT *C. neoformans* die rapidly compared to PBS-treated controls. Death events were recorded daily. Each infection condition represents survival of 95 animals, pooled together from 4 biological replicates, and 6 total technical replicates. Statistical significance was assessed by Gehan–Breslow–Wilcoxon test, which we used to place weight on early time points in the survival curve. We used this test because we expected to see the glyphosate-mediated differences early in the infection due to the timing of the glyphosate treatment. Since where the expected effects of the one-time pretreatment with glyphosate would be. (**D**) The size of the dark melanized particles within nodules upon *C. neoformans* infection are significantly smaller in the glyphosate-treated (10 μl of 1 mM) infected groups compared to the PBS-treated infected groups, which were analyzed for significance using a nested nonparametric Mann–Whitney–Wilcoxon rank test. Horizontal bar represents the median value and the error bars represent the 95% confidence interval (**E**) The degree of melanin encapsulation of the yeast within the nodule is also reduced in the glyphosate-treated (10 μl of 1 mM) groups, as measured on a scale of 0 (no melanin encapsulation) to 4 (very high levels of melanin encapsulation) as demonstrated in (**F**). Numbers in each bar represent the number of encapsulated *C. neoformans* for each score. Statistical significance was assessed using a chi-squared table test. Data in (**D**) and (**E**) represent data over 3 independent replicates with 3 larvae used per condition per replicate. (**G**) Representative brightfield micrographs showing the hemocyte and nodule formation in the different treatment groups at 20× and 100× magnification. Scale bars represent 10 μm. Nested nonparametric Mann–Whitney–Wilcoxon rank test performed using R for R 4.0.2 GUI 1.72 for Mac OS at https://www.r-project.org/ (R Core Team, 2020) and the *nestedRanksTest* package (*Version 0.2*, D.G. Scofield, 2014) [77]. All other statistical analyses performed using GraphPad Prism version 8.4.3 for Mac OS, GraphPad Software, San Diego, California, USA, www.graphpad.com. For underlying data, please see Data Availability section and/or S1 Table. See also S1 Fig. AMPA, aminomethylphosphonic acid; L-DOPA, 3,4-dihydroxyphenylalanine; PO, phenoloxidase; WT, wild-type.

To investigate whether glyphosate rendered mosquitoes more susceptible to infection with the human malaria parasite *P. falciparum*, adult female mosquitoes were fed on 10% sugar solution supplemented with glyphosate at different concentrations for 5 days and then given a *P. falciparum*-infected blood meal. Parasite burden was assessed through enumeration of the *Plasmodium* oocyst stage at 8 days postinfection (DPI). Glyphosate-fed mosquitoes had higher oocyst burdens with an overall nonsignificant trend of increasing oocyst burden with increasing dose of glyphosate (Fig 2B). However, we observed a sharp decline in parasite burden in the 10 mM-treated group, which was likely due to the increased mortality of this group (Fig 2C) resulting in few surviving mosquitoes to assess the intensity of infection. In a low *P. falciparum* infection intensity assay (S2A Fig), we observed that glyphosate-treated groups are more likely to be infected than control groups. This is important, as lower parasite burdens are more reminiscent of infections in field conditions in malaria-endemic regions [80–83].

Sugar preparations with glyphosate at environmentally relevant concentrations were given to *A. gambiae* mosquitoes to ascertain the herbicide's effect on the mosquito's life span. Compared to control mosquitoes (sugar-fed without glyphosate), mosquitoes given low glyphosate doses (30 to 300 μM) showed statistically significant improved survival, while those fed higher doses of glyphosate (1 to 10 mM) had equal or decreased survival, with the 10 mM glyphosate-exposed group exhibiting significantly decreased survival (Fig 2C). Additionally, we used a Cox Mixed Effects Model to account for fixed and random effects, and calculated the hazard ratios for each of the treatment groups (Fig 2D). Hazard ratios <1 indicate a reduced risk of death compared to the control, while hazard ratios >1 indicate enhanced risk of death compared to the control group. With this model, the mosquitoes treated with lower glyphosate concentrations had a hazard ratio less than 1, those treated with 1 and 3 mM glyphosate had hazard ratios similar to 1, and the 10 mM-treated mosquitoes had a hazard ratio significantly greater than 1. These results suggest that glyphosate could have bimodal effects on mosquito health.

We also measured the impact of glyphosate on the mosquito cuticle and body size. There was no discernable difference in *A. gambiae* cuticle pigmentation after 5 days of treatment with 1 mM glyphosate in 10% sucrose from days 3 to 8 postemergence, as measured by mean gray value of the ventral abdomen (S2C Fig). This was expected due to the typical expression of cuticular laccases being largely at the timing of pupation and the first 3 days postemergence [23,24]. Additionally, there were no difference in wing length, a proxy for adult size, between

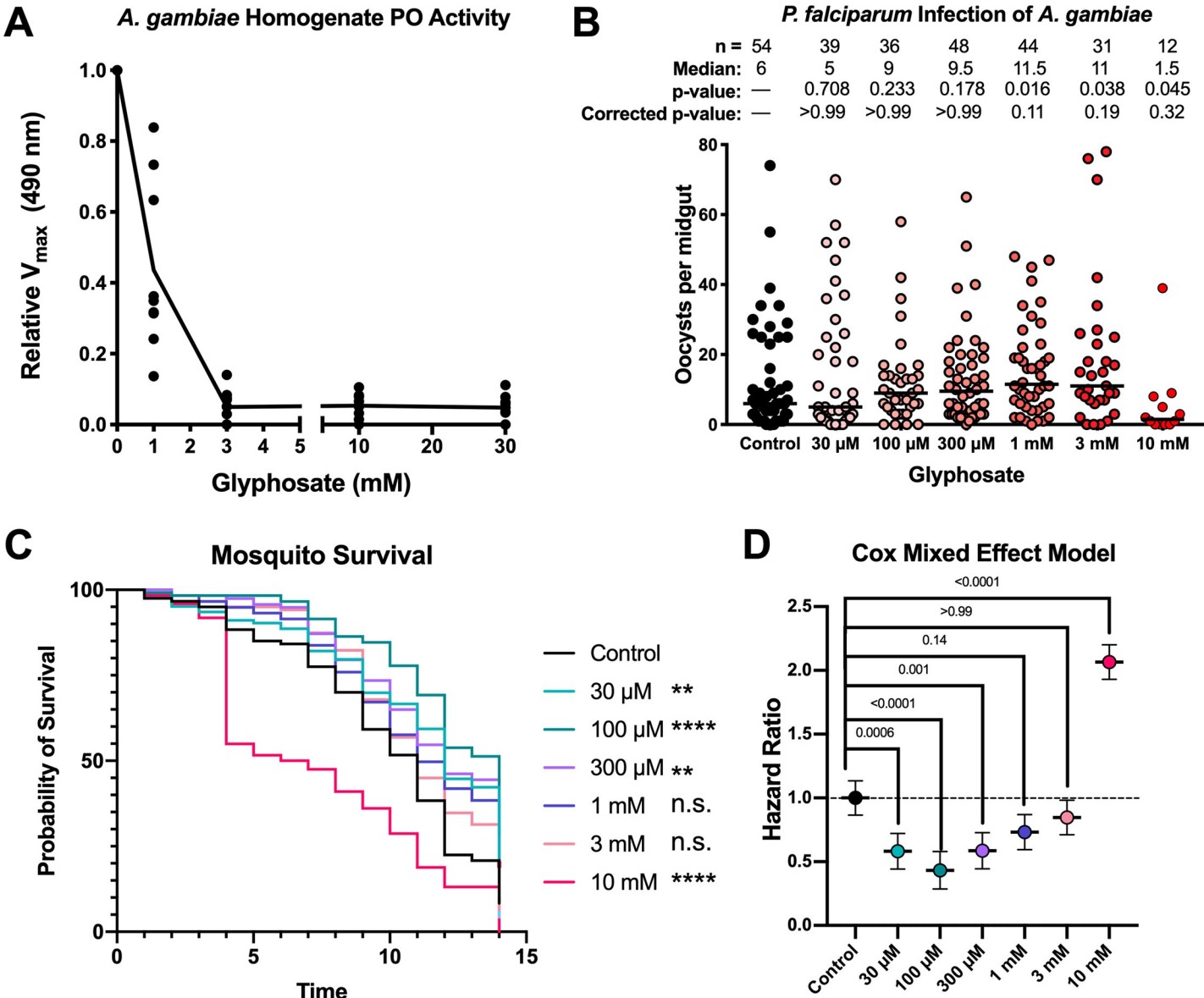

**Fig 2. Glyphosate effects on *A. gambiae* phenoloxidase activity and susceptibility to *Plasmodium* infection.** (A) Glyphosate inhibits phenoloxidase activity in *A. gambiae* homogenate. Enzyme activity represents 3 biological replicates with 3 technical replicates for each condition. (**B**) Glyphosate treatment increases the susceptibility of the *A. gambiae* to *P. falciparum* infection as measured by oocyst count per midgut. Increased glyphosate doses are associated with increased median oocyst burden. Parasite infection represents 4 biological replicates and 4 separate infections, line indicates median, and differences in parasite burden analyzed for significance using nonparametric Kruskal–Wallis test with each group compared to the control group with Dunn correction for multiple comparisons. (**C**) Low doses of glyphosate enhance the survival of adult mosquitoes, while the higher doses diminish their survival as compared to the control. Survival curves represent 120 animals from 3 independent replicates composed of groups of 40 mosquitoes, and survival was examined for statistical significance using the log-rank Mantel-Cox analysis with a Bonferoni correction for multiple comparisons. (D) Hazard ratios calculated from the Cox Mixed Effects Model to account for fixed (glyphosate treatment) and random effects (replicate). Hazard ratios <1 indicate lower risk of death compared to control values, and values >1 indicate a higher risk of death compared to the control. Hazard ratio of 1 is depicted by a dotted line. The Cox Mixed Effects modeling was performed using R for R 4.0.2 GUI 1.72 for Mac OS at https://www.r-project.org/ (R Core Team, 2020) and the *coxme* package (*Version 2.2–16*, T.M. Therneau, 2020) [79]. All other statistical analyses performed using GraphPad Prism version 8.4.3 for Mac OS, GraphPad Software, San Diego, California, USA, www.graphpad.com. The statistical significance in (**C**) is coded as: ns, $p > 0.05$; $^*p < 0.05$; $^{**}p < 0.01$; $^{***}p < 0.001$; and $^{****}p < 0.0001$. For underlying data, please see Data Availability section and/or S1 Table. See also S2 Fig. n.s., not significant; PO, phenoloxidase.

the glyphosate-treated and untreated adult mosquitoes (S2D Fig). Altogether, this suggests that the observed increased parasite burden in glyphosate-treated mosquitoes cannot be explained merely by broader impacts of glyphosate on mosquito health.

## Glyphosate alters the composition, but not density, of the *A. gambiae* midgut microbiota

*A. gambiae* midgut microbiota can influence *Plasmodium* infection by modulating the mosquito's innate immune system and hence affecting parasite viability [84–87]. We investigated whether glyphosate had detrimental effects or influence on the *A. gambiae* microbiota. Colony forming unit (CFU) counts from cultures of midgut homogenates grown on LB agar demonstrated that glyphosate treatment did not affect total number of culturable gut bacteria (Fig 3A), though this method would miss any impacts on microbes that were not readily cultured by these methods. To complement our culture-dependent analysis and provide insight on microbiota community composition, we compared the total 16S rRNA composition of the midgut microbiota with and without glyphosate treatment. Glyphosate treatment altered microbiota composition, with a noted decrease in the relative abundance of Enterobacteriaceae and an increase in relative *Asaia* spp. populations (Fig 3B). We did not observe a dose-dependent impact on composition, as the alpha diversity (a function of the number of bacterial taxa) of mosquitos exposed to glyphosate was similar to controls (Fig 3C). However, community composition was perturbed by treatment with glyphosate, and glyphosate-treated groups and controls form 2 separate clusters in principal coordinates analysis as measured by Bray–Curtis dissimilarity (Fig 3D). These differences suggest a shift in beta diversity (prevalence of each bacterial taxon), and therefore a difference between the microbial communities of mosquitoes exposed to glyphosate versus untreated controls.

## Glyphosate inhibits production of dopaquinone, dopachrome, and melanin

To understand how glyphosate inhibited melanization, we evaluated the formation of melanin intermediates in a stepwise manner using a commercially available fungal tyrosinase and the melanin precursor L-DOPA (2 mM). Although this tyrosinase differs from insect phenoloxidase, the melanization reaction in these systems follows the same Mason–Raper pathway (Fig 4A) [88,89] and thus can be used to explore the mechanism of glyphosate inhibition. The first step of the reaction involves L-DOPA oxidation to dopaquinone enzymatically or spontaneously [90]. We found that glyphosate inhibited the dopaquinone production in a dose-dependent manner (Fig 4B). This inhibition was observed for both tyrosinase-mediated and auto-oxidation-mediated production of dopaquinone. The slopes of inhibition in the auto-oxidation and tyrosinase-mediated oxidation were similar. This indicated that the tyrosinase reaction dopaquinone levels would remain unchanged by glyphosate treatment if the inhibition of "background" auto-oxidation dopaquinone production were taken into consideration. These results suggested that dopaquinone inhibition was primarily rooted in preventing the oxidation of L-DOPA independent of tyrosinase.

Dopaquinone spontaneously cyclizes to form cyclodopa, which then undergoes a redox exchange with another dopaquinone molecule to form 1 molecule of dopachrome and 1 reformed molecule of L-DOPA. Dopachrome is a pink-orange melanin intermediate that has an absorbance maximum at 475 nm. Dopachrome is a useful proxy product for tyrosinase-mediated reaction kinetics and evaluating the melanization reactions and redox exchange [91]. The rate of dopachrome formation and the amount of dopachrome produced were determined by measuring changes in absorbance during a reaction between L-DOPA and tyrosinase. There was a strong dose-dependent inhibition of dopachrome formation with glyphosate (Fig 4C), implying that the compound's inhibitory effects were upstream of dopachrome.

We tracked the reaction over 5 days to confirm inhibition of melanin synthesis. Glyphosate inhibited the production of a black pigment dose-dependently, as measured by the absorbance

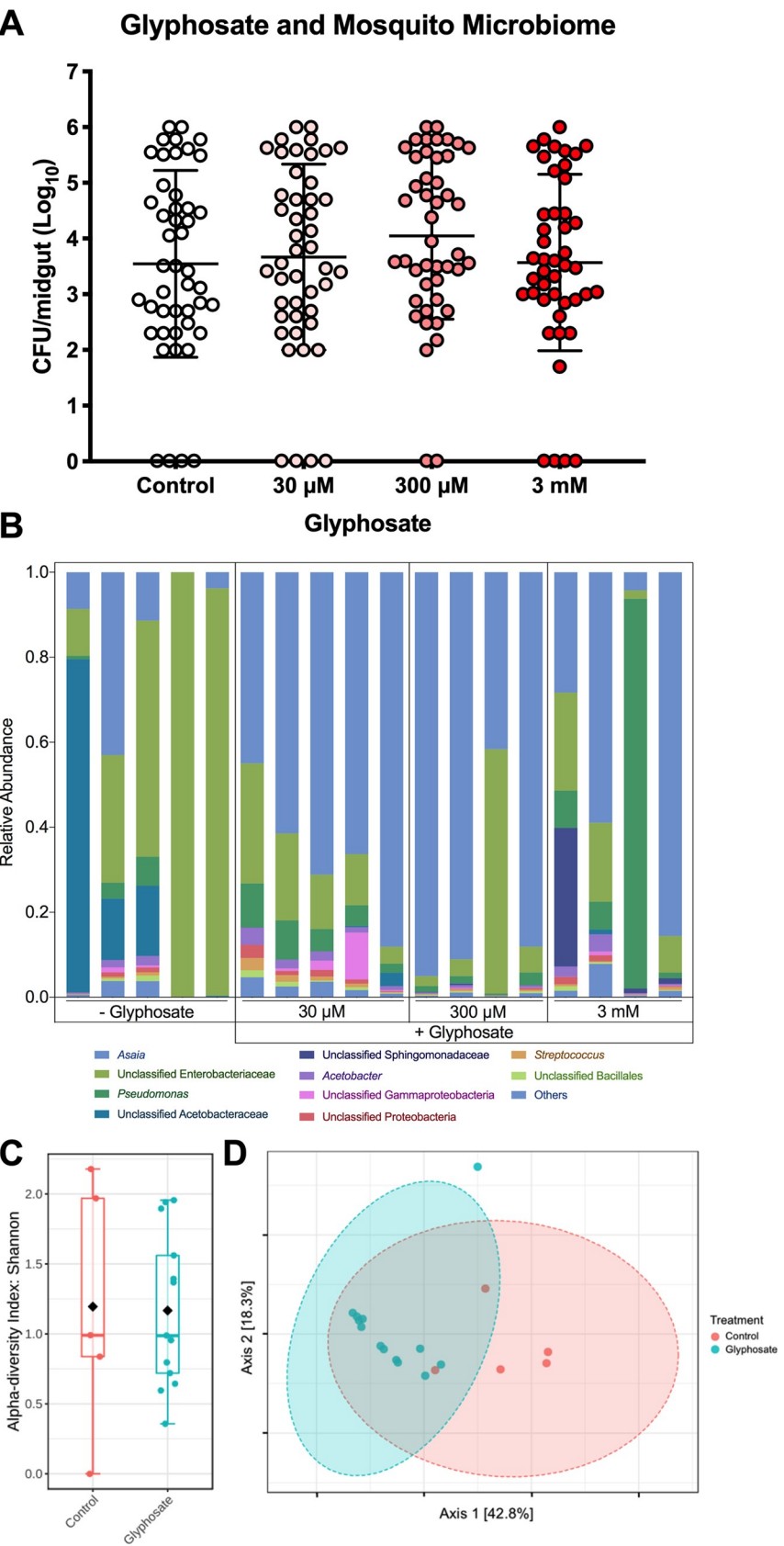

**A** **Glyphosate and Mosquito Microbiome**

**B**

Legend for B:

- *Asaia*
- Unclassified Enterobacteriaceae
- *Pseudomonas*
- Unclassified Acetobacteraceae
- Unclassified Sphingomonadaceae
- *Acetobacter*
- Unclassified Gammaproteobacteria
- Unclassified Proteobacteria
- *Streptococcus*
- Unclassified Bacillales
- Others

**C**

**D**

**Fig 3. Glyphosate alters the composition, but not density, of the *A. gambiae* midgut microbiota. (A)** Glyphosate does not alter microbial density of the culturable mosquito midgut bacteria (grown on LB agar). Each sample consists of 40–50 individual mosquito midguts over 3 independent replicates. Error bars represent the mean and ±SD. **(B)** Glyphosate alters the composition of the mosquito microbiota, leading to decrease of Entereobacteriacae and an increase of *Asaia* spp. **(C)** The glyphosate treatments do not significantly alter alpha diversity as measured by the Shannon Index (statistical analysis conducted using one-way ANOVA; NS = $p > 0.05$). **(D)** However, the glyphosate-treated and control-treated microbiota form distinct clusters in principle coordinates analysis, measured by Bray–Curtis dissimilarity. Statistical significance was tested by PERMANOVA ($p < 0.001$, R = 0.557). Each treatment group represents 5 individual mosquito midguts. For underlying data, please see Data Availability section and/or S1 Table. For more information, see also S3 Fig. CFU, colony forming unit; n.s., not significant.

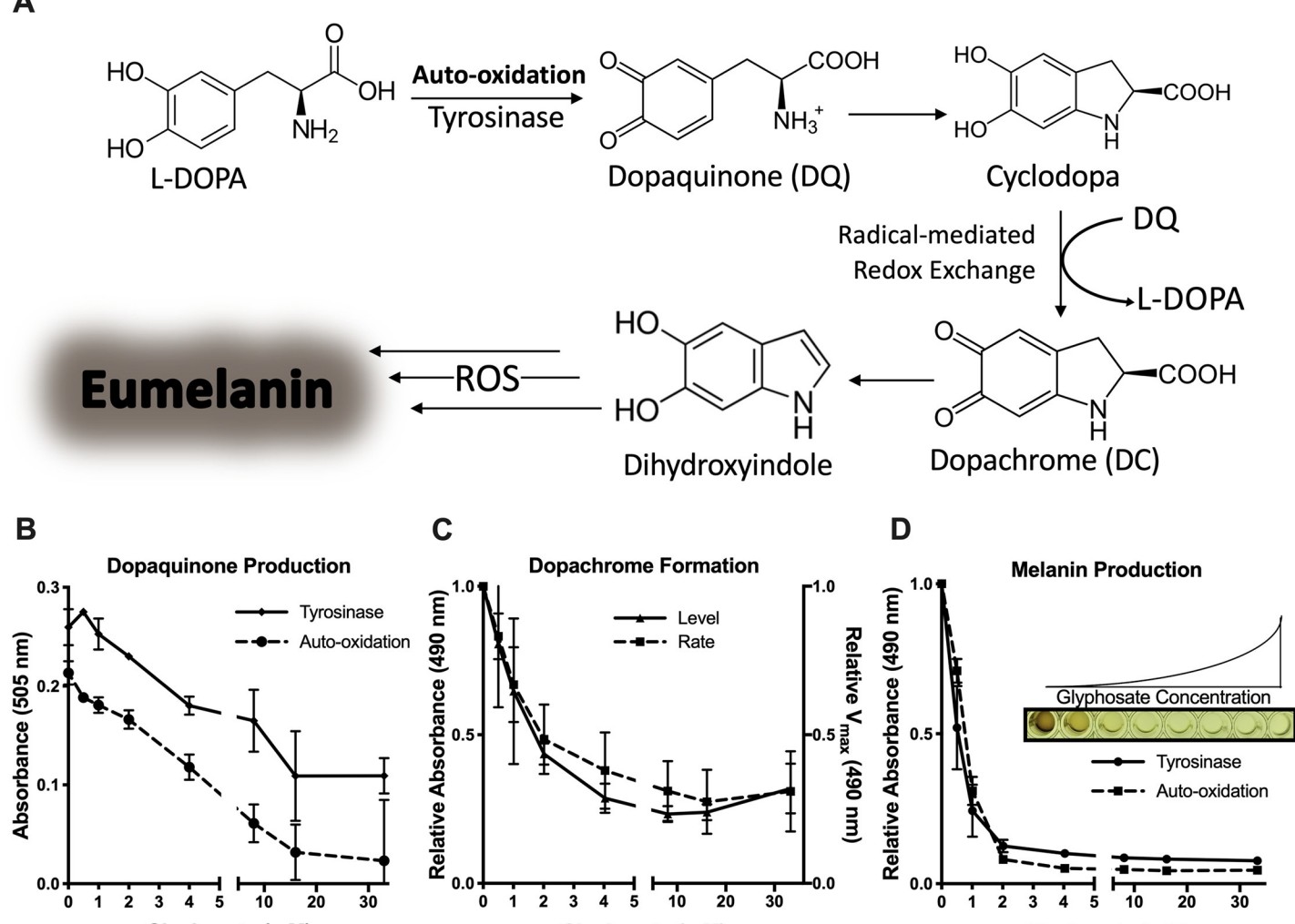

**Fig 4. Glyphosate inhibits in vitro melanin production. (A)** An overall schematic of the Mason–Raper pathway of melanization mediated by tyrosinase and auto-oxidation. (**B**) Glyphosate inhibits formation of dopaquinone produced by tyrosinase-mediated and spontaneous oxidation of L-DOPA. Dopaquinone is indicated by the absorbance of an MBTH-dopaquinone adduct pigment at 505 nm. Absorption levels are shown relative to the no glyphosate control with background (MBTH mixture) subtracted after 1 hour at 30˚C (**C**) Glyphosate decreases the rate of dopachrome formation and inhibits dopachrome production from tyrosinase oxidation of L-DOPA. Rate of dopachrome formation is the reaction $V_{max}$ at 490 nm relative to the $V_{max}$ without glyphosate. Dopachrome production is shown as the absorbance at 490 nm relative to the control after 30 minutes of reaction. (**D**) Melanin production is inhibited by glyphosate with tyrosinase and auto-oxidation of L-DOPA. Melanin levels are measured as the absorbance at 490 nm after 5 days of reaction. Inset shows a representative image of the data in (**D**) showing melanization inhibition with increasing glyphosate concentration. Values are depicted relative to the no glyphosate control. Error bars represent ±SD. Each experiment was performed at least 3 independent replicates. For underlying data, please see Data Availability section and/or S1 Table. L-DOPA, 3,4-dihydroxyphenylalanine; MBTH, 3-methyl-2-benzothiazoninone hydrazone; ROS, reactive oxygen species.

of the tyrosinase reaction on day 5 (Fig 4D). Interestingly, glyphosate also inhibited the formation of pigment that derives from auto-oxidation of L-DOPA (Fig 4D). This implies that glyphosate inhibited pigment production nonenzymatically.

## Phosphate-containing compounds inhibited melanization similarly to glyphosate

To gain insight into the chemical features of glyphosate that inhibited melanogenesis, we assayed several structurally similar compounds using the same in vitro mushroom tyrosinase assay. To test the effect of the amino acid functional group, we compared glyphosate alongside its non-phosphate analog, glycine. We also tested the inhibitory effects of phosphoserine and serine on melanin production. Phosphoserine inhibited dopaquinone, dopachrome, and melanin formation to nearly the same extent as glyphosate (Fig 5A–5C). In contrast, neither glycine nor serine inhibited dopaquinone, dopachrome, or overall melanin formation (Fig 5A–5C). We tested the inhibitory effects of other phosphate-containing compounds including organophosphates (phosphonoacetic acid), phosphoesters (pyrophosphate), and phosphoric acid. All of the phosphate-containing compounds inhibited dopaquinone production (Fig 5A) and dopachrome formation (Fig 5B) in a manner nearly identical to glyphosate, but differed slightly from each other in melanin inhibition (Fig 5C).

Similar to glyphosate, these compounds all inhibited auto-oxidation of L-DOPA comparably to their inhibition of tyrosinase-mediated melanin production (Fig 5E). This further illustrates that glyphosate and similar phosphate-containing compounds inhibit melanin in a nonenzymatic fashion. These data suggest that the phosphate functional groups of these compounds may be responsible for the melanin-inhibitory properties.

## Glyphosate does not react with L-DOPA or inhibit tyrosinase directly

We considered the possibility that glyphosate inhibited melanogenesis and dopaquinone production by reacting with the L-DOPA substrate. To measure the reaction between these compounds, we analyzed mixtures of L-DOPA and glyphosate by $^{1}$H-NMR and $^{31}$P-NMR. We found no evidence of interaction between the 2 compounds based on peak shifts of hydrogen and phosphorous at both high (60 mM glyphosate and 20 mM L-DOPA) and low concentrations (6 mM glyphosate and 5 mM L-DOPA) (S4 Fig).

If glyphosate was inhibiting melanin production through the formation of a covalent bond with tyrosinase, the inhibition should be irreversible. To test this, we treated 20 µg/ml tyrosinase with 5.63 mg/ml (33.33 mM) glyphosate and removed the glyphosate by dialysis. The glyphosate-treated enzyme had similar activity to the control (Fig 6A), making a strong case against a mechanism whereby glyphosate inhibited melanogenesis through irreversible inhibition of tyrosinase. Instead, analysis of the tyrosinase reaction by Michaelis–Menten kinetics assay with L-DOPA and glyphosate suggested that glyphosate is a noncompetitive inhibitor of melanin and dopachrome production (Fig 6B). Further, we tested tyrosinase activity as a function of enzyme concentration with and without glyphosate and constant concentration of L-DOPA. We found that the slope of the glyphosate-treated enzyme is lower than the water-treated control (Fig 6C). This indicates that glyphosate-mediated inhibition is reversible [92,93]. Given that our findings showed that glyphosate inhibits auto-oxidation and tyrosinase-mediated oxidation, we believe that the reversible inhibition is due to glyphosate interfering with the L-DOPA substrate's ability to be oxidized rather than the enzyme's ability to oxidize. This could be represented by the following where *E* represents tyrosinase, *S* represents

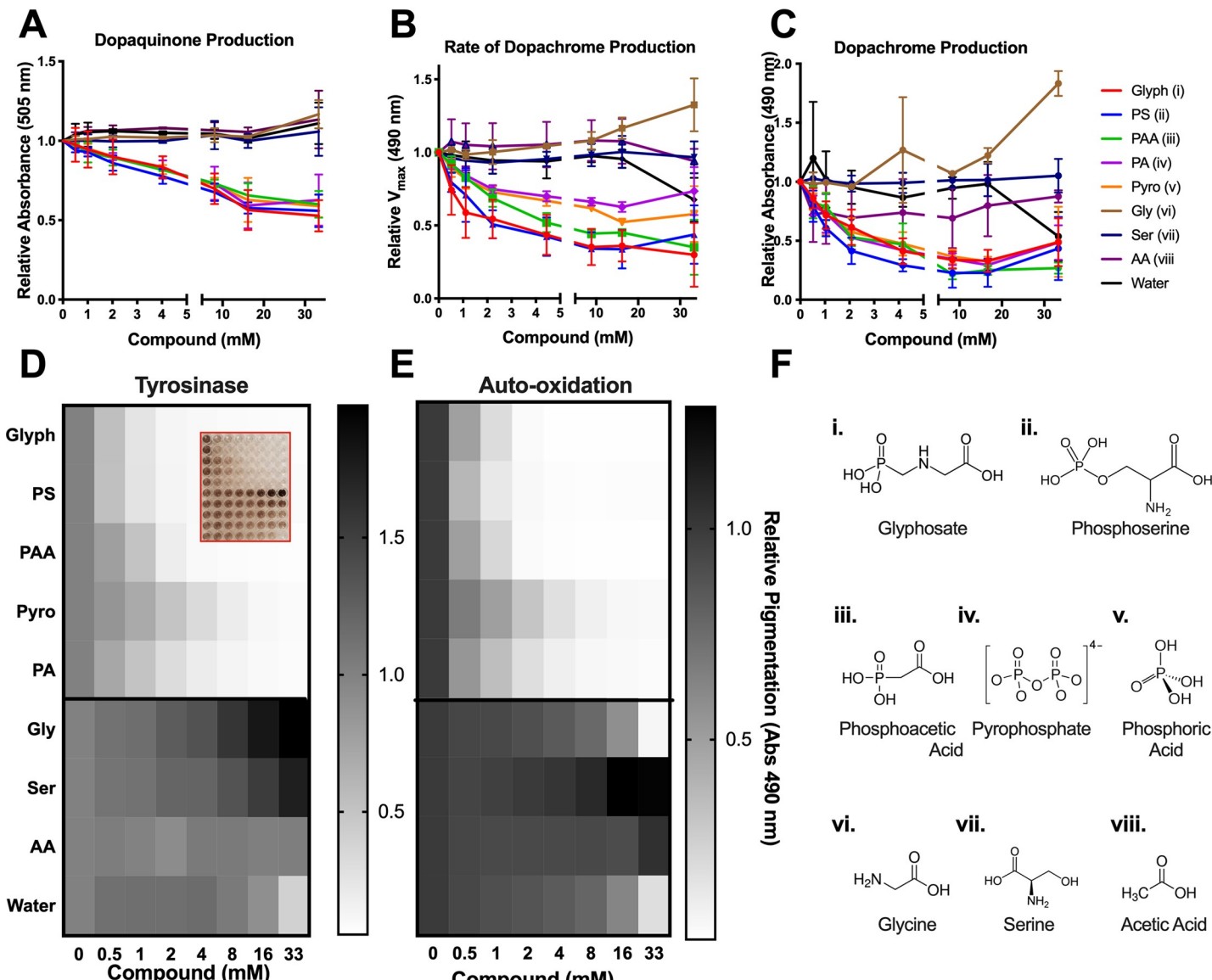

**Fig 5. Phosphate-containing compounds inhibited melanization similarly to glyphosate.** Glyphosate, PS, PAA, pyro, and PA inhibit dopaquinone formation (**A**), rate of dopachrome formation (**B**) and dopachrome levels (**C**), and melanin formation (**D**), whereas their respective non-phosphate analogs, gly, ser, and AA do not inhibit any step of melanization (**A–D**). (**E**) Auto-oxidation of L-DOPA is inhibited by glyphosate, PS, PAA, pyro, and PA in a similar manner. The compounds tested (**F**) were diluted in 300 mM stock solution and titrated to pH between 5 and 6. Absorption and rates are shown relative to the internal no drug control. Grayscale bars represent mean absorbance at 490 nm relative to no compound control. The darker colors correspond to increased pigment formation. Inset shows a representative image of the data in (**D**) showing the effects of the compounds on melanization. Error bars in (**A–C**) represent ±SD. Each experiment represents at least 3 independent replicates. For underlying data, please see Data Availability section and/or S1 Table. AA, acetic acid; gly, glycine; L-DOPA, 3,4-dihydroxyphenylalanine; PA, phosphoric acid; PAA, phosphonoacetic acid; PS, *o*-phosphoserine; pyro, pyrophosphate; ser, serine.

L-DOPA, $I$ represents glyphosate, and $P$ represents dopaquinone/melanin:

$$\text{Normal Enzymatic Reaction}: \; E + S \rightleftharpoons ES \longrightarrow P$$

$$\text{Inhibited Enzymatic Reaction}: \; E + S + I \rightleftharpoons E + SI \dashv P$$

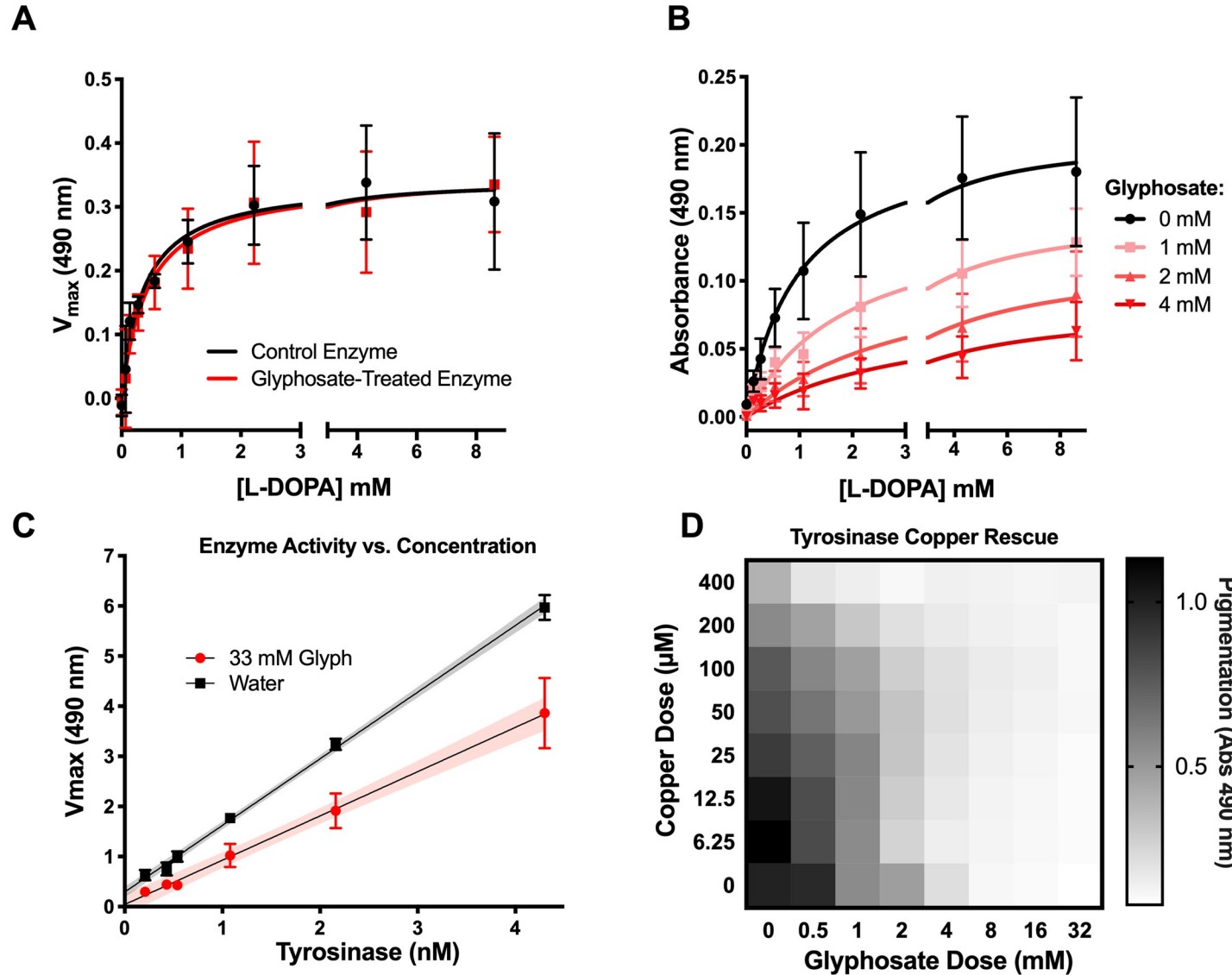

**Fig 6. Glyphosate does not directly inhibit tyrosinase activity. (A).** Tyrosinase activity is not irreversibly inhibited, and glyphosate-treated enzyme has normal activity when glyphosate is dialyzed out of solution. **(B)** Glyphosate appears as a noncompetitive inhibitor of tyrosinase in Michaelis–Menten kinetics assays measuring the change in absorbance at 490 nm over 24 hours compared to the no tyrosinase background. **(C)** The rate of dopachrome formation with glyphosate treatment is smaller than the slope of the control treatment across all concentrations of tyrosinase. This reduced slope indicates reversible inhibition. The assay is performed under constant L-DOPA and glyphosate concentrations. Shaded areas represent the 95% CI of the linear regression. **(D)** Adding $Cu^{+2}$ to L-DOPA-tyrosinase reactions with glyphosate does not rescue melanin inhibition compared to the glyphosate-free control. (See also S5 Fig) Grayscale bars represent mean absorbance at 490 nm relative to no glyphosate and no copper control. The darker colors correspond to increased pigment formation. Error bars in **(A–C)** represent ±SD. Each experiment represents at least 3 independent replicates. For underlying data, please see Data Availability section and/or S1 Table. L-DOPA, 3,4-dihydroxyphenylalanine.

Copper ions are important for tyrosinase activity. Since glyphosate is a metal chelator, [94,95], we evaluated whether glyphosate's inhibitory effect was due to this property. We added copper ions to the L-DOPA and tyrosinase reaction to rescue the glyphosate inhibition. We performed the experiment with 8 concentrations of copper (II) sulfate for each of the 8 glyphosate concentrations. In general, the addition of copper did not rescue the glyphosate dependent inhibition of melanin (Fig 6D). However, low concentrations of copper (6.25 to 25 μM) increased tyrosinase activity, while high concentrations of copper (50 to 400 μM) reduced activity, indicating low copper can boost enzyme activity, while higher concentrations

inhibit the reaction. However, this hormesis-like effect was not observed at increasing glyphosate concentrations (S5 Fig). This result indicates that glyphosate's ability to chelate copper ions could have a protective effect in high copper environments, which would otherwise lead to negative effects on enzymatic activity and other biological processes. Similar results have been previously seen in *Eisenia fetida* earthworms exposed to high copper conditions in soil [96]. Glyphosate contamination of copper-rich soil reduced the detrimental effects of the metal's toxicity, presumably due to the glyphosate's copper chelation properties [96].

## Glyphosate affects the oxidative properties of melanogenesis

Melanogenesis is dependent on the spontaneous radicalization of quinone intermediates [97]. Dopaquinone radicals and cyclodopa undergo a radical-mediated redox exchange that converts cyclodopa into dopachrome and dopaquinone into L-DOPA. Further downstream, ROS catalyze the polymerization of dihydroxyindole into eumelanin. Glyphosate's inhibitory effect could be due to a role as a free-radical scavenger or antioxidant. Since the inhibitory compounds blocked spontaneous oxidation of L-DOPA (Fig 5E), they are antioxidants. To measure this radical-quenching ability, we used a 2,2′-azino-bis(3-ethylbenzothiazoline-6-sulfonic acid) (ABTS) assay in which ABTS radicals are blue, but when quenched, the solution becomes colorless. The degree of discoloration is a proxy for radical concentration and antioxidant strength. Glyphosate quenched the ABTS radical to some degree, but only after several hours of reaction (S6A Fig), which did not occur with the other inhibitory phosphate group containing compounds evaluated (Fig 7A). This indicates that direct free-radical scavenging may not be the primary mechanism of melanin inhibition for glyphosate.

Phosphoric acid is a well-known synergist that boosts the antioxidant properties of phenolic compounds. Phosphoric acid, and other synergists such as citric acid, malic acid, and tartaric acid do not directly quench free radicals themselves, but instead work by regenerating antioxidants, thus becoming "sacrificially oxidized," or chelating metal ions in solution [98,99]. Alternatively, glyphosate could be reacting with existing antioxidants to strengthen and/or regenerate them into "active" form. In this instance, the glyphosate would be bolstering the antioxidant properties of L-DOPA.

We observed that the synergist citric acid inhibited melanization similarly to glyphosate and phosphoric acid (Fig 7B and 7C). The addition of glyphosate, phosphoserine, and phosphoric acid enhanced the antioxidant properties of L-DOPA in an ABTS assay in a similar manner as citric acid (Fig 7D). This suggests that glyphosate may act as an inhibitor via antioxidant synergism. The synergy is the ratio of the quenching capacity of the L-DOPA and the compounds alone to the quenching capacity of L-DOPA combined with the compound. The lower this ratio, the more synergistic the compounds are with L-DOPA (S6B Fig). These values indicate that the inhibitory compounds are synergistic, whereas the noninhibitory glycine and serine are not as synergistic.

The inhibition of melanin was independent of the L-DOPA to glyphosate ratio, and glyphosate's $IC_{50}$ is approximately 1 mM regardless of L-DOPA concentration (S7 Fig). This could be explained by a general antioxidant effect on solution.

## Glyphosate alters the oxidation–reduction potential of the system

L-DOPA is a more effective antioxidant when it is oxidized or radicalized and has a better ability to form adducts with other radicals [100]. Since glyphosate is acting as a synergistic antioxidant, it may be driving L-DOPA oxidation and possibly radicalization in which L-DOPA scavenges radicals better. This has the potential to disrupt melanin synthesis by stopping the spontaneity of redox exchange and dopaquinone formation.

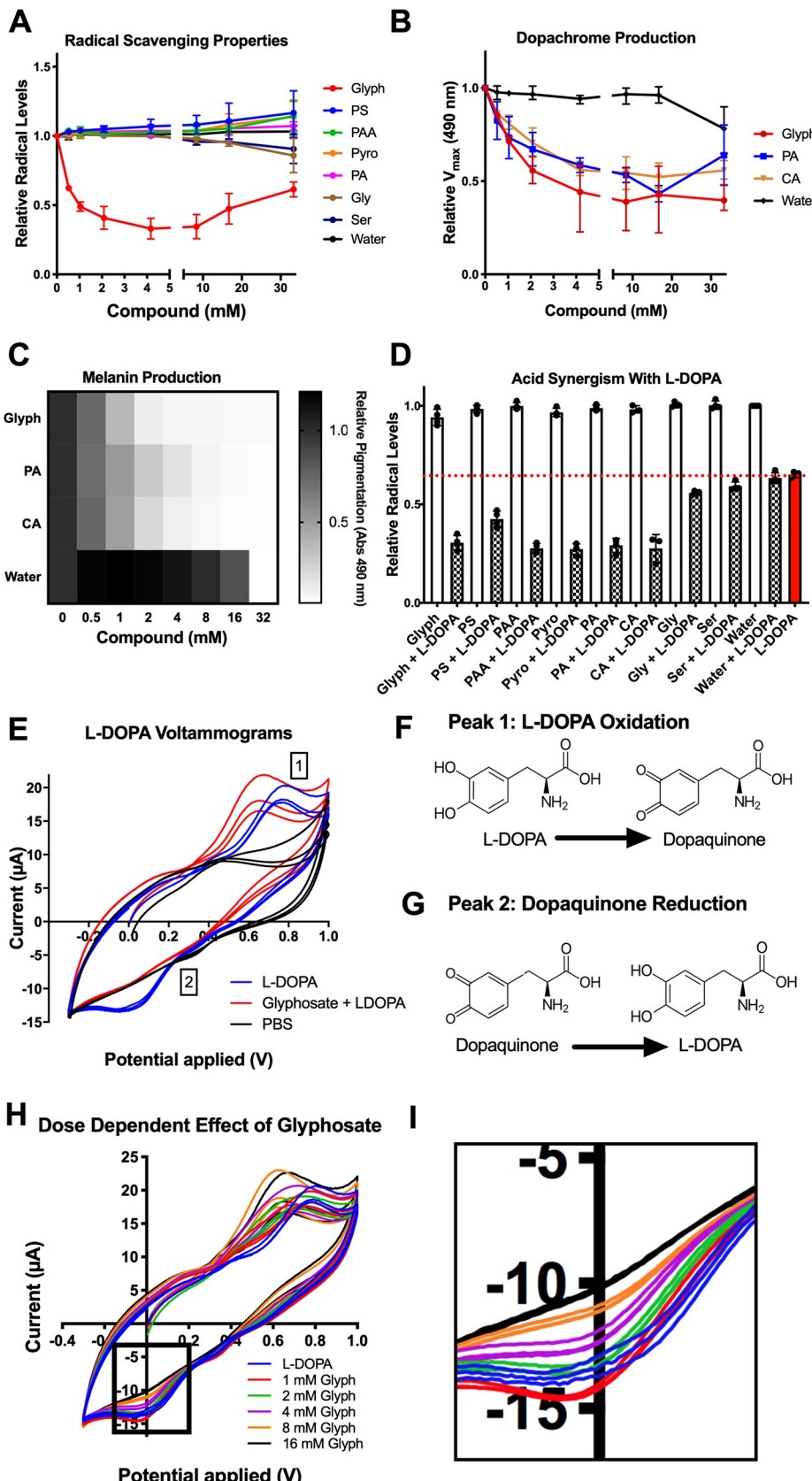

**Fig 7. Glyphosate affects the oxidative properties of melanogenesis.** (**A**) None of the melanin inhibitors exhibit radical quenching properties in an ABTS assay aside from glyphosate, which shows weak antioxidant properties after several hours in the ABTS solution. Absorbance at 734 nm is an indicator of how much ABTS remains in radical form (not quenched). (**B, C**) Citric acid (CA), a nonradical quenching antioxidant (antioxidant synergist), exhibits similar melanin inhibition as glyphosate and phosphoric acid, another known antioxidant synergist. Grayscale bars in (**C**) represent absorbance at 490 nm relative to no compound control, with the darker colors correspond to increased pigment formation. (**D**) Glyphosate, phosphoserine, phosphoric acid, and citric acid show synergy with the antioxidant L-DOPA. The addition of these compounds to L-DOPA enhances its radical quenching abilities by approximately 50%. Black dotted line represents the normalized ABTS absorbance treated with water. The other compounds tested here alone do not show much deviation from this line. The blue dotted line indicates the ABTS solution treated with L-DOPA alone. ABTS treated with L-DOPA and synergetic compounds together are below this line. (**E**) Average cyclic voltammogram showing the changes in oxidation and reduction of L-DOPA and dopaquinone when exposed to 16 mM glyphosate but not water. Numbers correspond to shifted peaks or peaks with less current compared to the water control. Peak 1 corresponds to L-DOPA oxidation (**F**); Peak 2 likely corresponds to dopaquinone reduction (**G**). Glyphosate shifts Peak 1 and 2 toward a decreased redox potential and diminishes the current of Peak 1 and 2 in a dose-dependent manner (**H**), notably decreasing Peak 2 current intensity to the point of nonexistence (**I**). Each experiment represents at least 3 independent replicates. Error bars in (**A, B, D**) represent ±SD. For underlying data, please see Data Availability section and/or S1 Table. See also S6 and S8 Figs. CA, citric acid; Glyph, glyphosate; L-DOPA, 3,4-dihydroxyphenylalanine; PA, phosphoric acid; PAA, phosphonoacetic acid; PS, *o*-phosphoserine; Pyro, pyrophosphate; Ser, serine.

To investigate whether the addition of glyphosate changed the oxidation properties of L-DOPA, we used cyclic voltammetry—a technique to measure the electrochemical properties of solutions and previously used to study quinone electrochemistry [101,102]. Voltammetry performed on L-DOPA solutions with glyphosate showed dose-dependent shifts towards a negative potential (Fig 7E and 7H) in peaks that corresponded to L-DOPA oxidation [103] (**Peak 1**). We validated these as L-DOPA oxidation peaks by performing voltammetry on various L-DOPA concentrations (S8A Fig). The peak shift towards negative potentials indicates the L-DOPA was oxidized more easily and had less ability to be an oxidant, similar to the negative potential shifts associated with alkaline pH and increased oxidation [104]. We controlled for any pH-dependent peak shifts by adjusting each solution to pH 6.00 prior to measurement. Decreased oxidizing power can lead to significant effects, as melanin biosynthesis is reliant upon catechol oxidation and high redox potentials of quinones. Interestingly, with increased glyphosate, the L-DOPA solution had a lower current intensity associated with the reduction of dopaquinone to L-DOPA (**Peak 2**). In cyclic voltammetry, smaller peaks indicate that less of the compound is oxidized or reduced. The decreased **Peak 2** current became virtually nonexistent with increasing glyphosate concentrations (Fig 7E, 7H, and 7I). This implies that dopaquinone, represented by **Peak 2**, is either not being formed during L-DOPA oxidation or cannot be reduced back into L-DOPA. These data indicate that the redox cycling steps of melanization are halted due to the inability of dopaquinone to be reduced into L-DOPA. This could also indicate that while L-DOPA was oxidized more in the presence of glyphosate, it may form a non-dopaquinone product—either a radical-mediated dimer with itself or a semiquinone.

## Discussion

We investigated the effect of glyphosate on melanin production in 2 species of insects, *G. mellonella* and *A. gambiae*, and found that both glyphosate and its major metabolite AMPA were inhibitors of insect phenoloxidase and melanization. Although glyphosate and AMPA are relatively weak inhibitors of these insects' melanization, the inhibitory concentrations are relevant in the environment given the vast amounts used in agriculture, their environmental stabilities, and the high potential for insect–herbicide interactions. Therefore, glyphosate has a high potential to influence key insect physiological systems. We observed that glyphosate enhanced the susceptibility to infection of 2 phylogenetically distinct insects, *G. mellonella* and *A. gambiae*. This raises concerns and the suggestion that glyphosate may interfere broadly with insect

immunity through its effects on melanin-based defenses. Analysis of in vitro tyrosinase and auto-oxidation models revealed that glyphosate inhibited melanization by acting as a synergistic antioxidant and disrupting redox cycling. Overall, our findings provide new insights on the complex reaction and suggest potential harmful effects of this herbicide on nontarget organisms, including some insects that may be important to ecosystem stability, and already in peril due to the threat of an "insect apocalypse."

*G. mellonella* treated with glyphosate were more susceptible to infection with *C. neoformans*. Glyphosate treatment was associated with reduced size of melanized nodules in the hemolymph following infection with *C. neoformans*. Two of three replicates showed significantly reduced numbers of melanized nodules in the glyphosate-treated infections. Nodules are primarily composed of hemocyte aggregates, released immune factors, and melanin encapsulation of the pathogen, which function together to kill invading pathogens [22]. Altogether, these data suggest that glyphosate weakened the melanin-based immune response of *G. mellonella*, which could have grave implications for host defense. *Galleria* are members of the order Lepidoptera (moths and butterflies), which represent up 10% of known species on Earth. Interactions with glyphosate in the soil, on plants during pollination, or ingested through herbivory could contribute to immunocompromised lepidopteran populations. Glyphosate's effects on immunity in insects could compound a controversial and preexisting problem of declines in Lepidopteran biomass in recent decades [25,105–108].

Like our observations with *G. mellonella*, glyphosate made the *A. gambiae* mosquito more susceptible to *P. falciparum* parasite infection, the primary agent of human malaria in Africa. However, melanization is not considered the primary anti-*P. falciparum* immune response in this malaria model [109]. The increased susceptibility of *A. gambiae* to *P. falciparum* could be due to broader alterations of mosquito immune defenses, or disruption of non-melanin roles of catecholamine oxidation and phenoloxidase in insect immunity including the production of ROS, cytotoxic intermediates, and pathogen lysis [3,13,110]. Importantly, we observed that even when infections of *A. gambiae* with *P. falciparum* resulted in an overall low to no parasite burden in control-treated groups, glyphosate-treated groups exhibited a higher infection burden and prevalence. This is notable because *Plasmodium* oocyte development within the mosquito is a major bottleneck to successful vector competence in nature [111]. If a mosquito can prevent oocyst formation, there is no transmission of malaria to humans. The numbers of oocysts from these low parasite burden experiments are in line with the normal burdens found in natural field infection models [112,113]. Our data may indicate that mosquitoes exposed to glyphosate were less able to control *Plasmodium* infection they would have otherwise resisted, thereby becoming potentially better vectors for malaria. Overall, our results raise concerns for public health and malaria control initiatives in regions in which malaria is endemic and where there is increasing use of glyphosate, including areas of Latin America, sub-Saharan Africa, and Asia.

Our data revealed that uninfected adult female mosquitoes treated with glyphosate displayed a hormesis-like dose-dependent effect when measuring survival outcomes. Survival increased at low doses of glyphosate compared to the control. This greater longevity may be due in part to reduced basal damage from host defense mechanisms that normally occur during melanin formation, and/or altered gut microbiota. In contrast, mosquitoes exposed to high concentrations of glyphosate showed decreased survival. These data suggest the broader notion that glyphosate could have varied and complex outcomes on vector competence depending on its concentration in the environment. The low-concentration glyphosate treatments resulted in longer-lived, yet immunosuppressed, mosquitoes that were slightly more susceptible to infection with *P. falciparum*, whereas short-lived high glyphosate-treated mosquitoes were much more susceptible to *P. falciparum*. Further, while the 10 mM-treated mosquitoes had the

worst survival outcome, the mosquitoes that survived the drugging showed low susceptibility to *P. falciparum* infection. These observations suggest a potentially interesting effect whereby very high concentrations of glyphosate reduce mosquito survival, but bolster the immune system or general physiology of survivors, which then allows them to resist *P. falciparum* infection with greater success. Alternatively, very high glyphosate treatment could be selecting for mosquitoes within the population more resistant to *P. falciparum* infection.

Our analyses of *A. gambiae* midgut microbiota indicated that glyphosate did not impact *A. gambiae* midgut culturable bacterial density, although the herbicide did perturb midgut microbiota composition in a non–dose-dependent manner. More specifically, glyphosate altered diversity of the microbial community, and glyphosate-treated mosquitoes exhibited diminished Enterobacteriaceae and expanded *Asaia* spp. populations. The presence of some Enterobacteriacae, including the common insectary contaminant *Serratia marcescens*, in *Anopheles* spp. midguts is associated with lower susceptibility to *Plasmodium* spp. infection [114,115]. This effect is observed quantitatively by the significantly different prevalence of individual bacterial taxons (beta diversity) between the glyphosate and control-treated microbiota, while there is an overall unchanged number of bacterial taxa present (alpha diversity). Beta diversity analysis indicates that microbial communities associated with glyphosate-treated mosquitoes cluster together and are different than those from control mosquito communities.

Our results are consistent with reports that glyphosate perturbs the microbiota of honeybees that makes them more susceptible to infection [59]. Our data suggest that while glyphosate may perturb the microbiota and affect immunity as previously described [59,60], it can also inhibit melanization which is a critical part of insect immune defense. We do not see a dose-dependent effect of glyphosate on the microbiota composition, but we do see a dose-dependent effect on mosquito susceptibility to *Plasmodium* infection; this indicates that the enhanced susceptibility might be unrelated to microbiome perturbations. These mechanisms of susceptibility are not mutually exclusive and could be additive to weaken insect health. Additionally, while AMPA does not disrupt the microbiota of honeybees [60], we show it can inhibit melanization of *G. mellonella* phenoloxidase and mushroom tyrosinase. A recent study in *Apis cerana cerana* honeybees indicate that glyphosate-based herbicide treatment increases expression of wound and defense genes, including those related to melanization [116]. Interestingly, this study also showed that glyphosate feeding decreased the expression of many odorant binding proteins, which have been shown to mediate the melanization response in both tsetse fly (*Glossina morsitans morsitans*) and *Drosophila melanogaster* [117] This suggests complex regulation of melanization following treatment with glyphosate-based herbicide, including the possibility of increased melanin-related gene expression as a compensation for glyphosate's inhibitory effects. Additionally, other surfactants and components of the commercial herbicide formulation used could trigger damage and immune gene expression.

Melanins and phenoloxidases are involved in other physiological functions in insects including proper pupation and cuticle and eggshell development. In our experiments, which involved single dosing or short duration of feeding, we did not detect a difference in coloration of adult mosquito treated with glyphosate, nor a defect in *G. mellonella* pupation following glyphosate treatments of final instar larvae. However, we cannot rule out that longer glyphosate exposure or feeding throughout the lifecycle would affect these functions. If such effects happen, they will only compound the effects of glyphosate on melanin-based immunity and insect physiology.

We sought to understand the mechanism of melanization inhibition by glyphosate. The process of melanization is highly dependent upon oxidation and redox cycling between catechols and quinones. The melanin production process is halted if the oxidizing ability and the redox potentials are altered. Melanization begins with the conversion of L-DOPA into

dopaquinone through enzymatic or spontaneous oxidation of L-DOPA, followed by redox cycling that results in dopachrome formation and, subsequently, melanin polymerization. Glyphosate inhibited formation of dopaquinone and melanin pigment mediated by both tyrosinase and auto-oxidation, which strongly suggests that glyphosate inhibits L-DOPA oxidation in an enzyme-independent manner. We found that other phosphate-containing compounds inhibited melanization in a similar manner including phosphoserine, phosphoacetic acid, pyrophosphate, and phosphoric acid. This is in line with literature reports that other aminophosphonic acids inhibit fungal eumelanin in the human pathogen *Aspergilius flavus* [118]. Incidentally, this class of compounds is patented for use in human cosmetics and is marketed as solutions to inhibit melanogenesis in the skin [119,120].

We found no evidence that glyphosate irreversibly inhibited tyrosinase activity or directly interfered with enzyme function. Addition of copper ions did not rescue the inhibition, indicating that the copper-based catalytic core of tyrosinase is not disrupted by glyphosate. Interestingly, low copper increased tyrosinase activity and high doses reduced activity. However, copper had minimal effects on tyrosinase activity during high glyphosate concentrations. It appears that glyphosate, possibly through chelation, acts as a "buffer" of copper ions and can reduce the metal's harmful effects, similar to previous findings concerning the toxicity of high-copper soil to earthworms following glyphosate treatment [96]. This could have broader implications for melanogenesis in nature, where some fungi use copper as a signal to up-regulate melanin-producing enzymes [121], and thus copper ion sequestration could reduce melanin production.

We examined the ability of glyphosate and the other compounds to quench free radicals, which are necessary to the melanization process. Of the inhibitors tested, only glyphosate had radical-quenching activity, but this occurred relatively slowly compared to the typical time-frame of antioxidant reactions reported in literature [122]. This property is likely not the mechanism of inhibition as phosphoserine has a similar structure and near identical inhibition of melanization as glyphosate, yet no radical-quenching properties. While not a free-radical quencher, phosphoric acid is a known antioxidant synergist—a class of compounds that enhance antioxidant properties of phenolic compounds by chelating metals or reverting antioxidants into their active states [99]. Synergists like phosphoric acid, citric acid, malic acid, and alpha-hydroxy acids are added to foods, medicines, and cosmetics at concentrations up to 10% as a preservative due to their synergist effects on antioxidants [123]. Glyphosate behaved similarly to phosphoric acid and citric acid; citric acid inhibited melanization similarly to glyphosate and phosphoric acid, suggesting an inhibition mechanism via antioxidant synergy. Additionally, we report that glyphosate and other inhibitors have synergistic effects on the antioxidant properties of L-DOPA. L-DOPA's antioxidant properties derive from its reduction back to a normal state from an oxidized state, or a radical-mediated adduction reaction with the oxidized compounds in solution. Since glyphosate makes L-DOPA a more efficient antioxidant, glyphosate thus alters the oxidative balance of L-DOPA and/or produces a buildup of radical or semiquinone intermediates.

Consistent with these findings of antioxidant synergy, cyclic voltammetry revealed that glyphosate decreased the L-DOPA-Dopaquinone redox potential. Hence, L-DOPA becomes both a weaker oxidizing agent and a stronger reducing agent (antioxidant) and is more prone to oxidation in the presence of the herbicide. Glyphosate decreased dopaquinone reduction in a dose-dependent fashion indicating that dopaquinone cannot be reduced or is not produced following L-DOPA oxidation. A lack of dopaquinone could indicate that glyphosate causes oxidized L-DOPA semiquinone intermediates to remain stable or react with each other and form L-DOPA dimers. On the other hand, if dopaquinone cannot be reduced into L-DOPA, melaninization becomes unfavorable as redox exchange could not occur. These changes in

voltammogram do not appear when the L-DOPA solution is treated with 16 mM glycine but did occur with citric acid. This further supports that glyphosate is acting as a synergistic anti-oxidant and prevents the redox-dependent melaninization.

Our findings investigating glyphosate's mechanism of melanin inhibition points to disruption of oxidative balance and redox cycling which may result in the buildup of toxic oxidative intermediates. Previous studies evaluating glyphosate's impact on organisms show that the herbicide increases oxidative stress, lipid peroxidation, and antioxidant responses in bacteria, plants, arthropods, fish, amphibians, rats, and human red blood cells [69–76]. These data bolster our findings that glyphosate promotes oxidation in phenolic compounds like L-DOPA and inhibits clearance of oxidative stress. Understanding the mechanisms by which compounds such as glyphosate might impact insect biomass and contribute to a potential insect decline is important, as they have both direct and indirect impacts on human health.

Glyphosate's interference with melanization could have considerable environmental impact given its stability and wide concentration range, from over 50 mM at time and at site of application to under 1 nM in runoffs from application sites [33,39,124]. At higher concentrations, glyphosate could inhibit melanin production in some insects, thus rendering them more susceptible to pathogens due to reduced immune competence. This suggests protean consequences for human health ranging from ecosystem disruption to altered vector competency of lethal human pathogens and increased malaria transmission in endemic regions that use glyphosate-based herbicides in agriculture. Importantly, we provide evidence that glyphosate enhances *A. gambiae* susceptibility to the human malaria parasite, which could potentially make it a better vector for transmitting disease to humans. Our data in *Galleria* and *Anopheles* can perhaps be extrapolated to other lepidopteran (moth and butterfly) and dipteran (fly) species with additional importance to the environment.

In summary, our results suggest that glyphosate interferes with melanization in 2 insect species, through a mechanism involving altering the redox potential of melanin polymerization reaction. This phenomenon is concerning because of the importance of melanization in insect immunity. A strong immune response is vital for insect survival, and disruption of their immune function, including the inhibition of melanization, could be disastrous for these animals. Insects are pivotal members of the world's ecosystems, essential to maintaining proper function, and they ensure human food security. Yet, certain data indicate a drop in insect biomass over recent decades, a phenomenon that has been called the "insect apocalypse" [25,105,106,125,126]. Although this view has been questioned regarding the true extent and possible causes of the insect population declines [127–131], our results suggest that glyphosate use as a mechanism by which insect immunity can be undermined by human activities.

## Methods

### Biological materials

*G. mellonella* larvae were obtained through Vanderhorst Wholesale, St. Marys, Ohio, USA. *C. neoformans* strain H99 (serotype A) was kept frozen in 20% glycerol stocks and subcultured into Sabouraud dextrose broth for 48 hours at 30˚C prior to each experiment. The yeast cells were washed twice with PBS, counted using a hemocytometer (Corning, New York, USA), and adjusted to $10^6$ cells/ml.

*A. gambiae* (Keele strain) mosquitoes were maintained on sugar solution at 27˚C and 70% humidity with a 12-hour light to dark cycle according to standard rearing condition. *P. falciparum* NF54 (Walter Reed National Military Medical Center, Bethesda) infectious gametocyte cultures were provided by the Johns Hopkins Malaria Research Institute Parasite Core Facility

and were diluted to 0.05% gametocytemia with naïve human blood before feeding to the mosquitoes using an artificial glass membrane feeder as established in [84].

## Compound and dilution preparation

Each compound, including the glyphosate (Millipore Sigma, Product #45521), was prepared in 300 mM stock solution in Milli-water Q and brought to a pH of 5.5, and 20 µl of each compound was serially diluted 1:2 in PBS (pH 7.4), with a compound-free control. When all reaction components are added, the final concentrations of the compounds in these dilutions were 33.33, 16.67, 8.33, 4.17, 2.08, 1.04, 0.52, and 0 mM.

## *Galleria mellonella* hemolymph extraction and phenol oxidase activity

Healthy (active and cream-colored) larvae were cold anesthetized and punctured in their proleg with 18G needle, and pressure was applied to the larvae to promote bleeding of hemolymph. Hemolymph was collected from larvae directly into an eppendorf tube. Anticoagulants were not used as they might interfere with the melanization process.

For automelanization experiments, hemolymph was diluted 1:10 in PBS and mixed with a pipette. Then, 160 µl of 1:10 hemolymph is added to 20 µl of glyphosate serially diluted in PBS. The change in absorbance at 490 nm was recorded and analyzed as described above.

For experiments with L-DOPA, hemolymph was diluted 1:5 in PBS and mixed by pipette. Experiments were performed as per the phenoloxidase activity assay in [132].

To test the effect of glyphosate on hemocytes viability, hemolymph was diluted 1:2 with anticoagulation buffer [133], as melanization was not of importance for this experiment. Hemocytes were pelleted and suspended in anticoagulation buffer. Glyphosate was added to an aliquot of hemocytes in solution and incubated with mixing on a rocker at 30˚C for 15 minutes. Hemocyte viability was assessed by 0.02% trypan blue staining and enumeration of stained (dead) versus unstained (alive) hemocytes with a hemocytometer.

## *Galleria mellonella* infection and survival

Healthy final instar *G. mellonella* larvae weighing between 175 and 225 mg were selected and left starving overnight. Groups of larvae were injected with 10 µl of PBS or 10 µl of 1 mM sterile glyphosate in PBS. Larvae were monitored and left to recover for 5 hours. Larvae were then injected with 10 µl of sterile PBS or injected with $10^4$ *C. neoformans* yeast cells per larva. Due to the low concentration of glyphosate administered to the larvae, their volume of hemolymph, and their body volume, we believe the approximate concentration of glyphosate is below the concentrations required to inhibit *C. neoformans* growth [26]. *G. mellonella* larvae and pupae were kept at 30˚C and monitored daily for survival for 14 days. Survival was assessed by movement upon stimulus with a pipette. See S9A Fig.

## Melanization and nodule measurements

*G. mellonella* larvae were drugged and infected as described above in groups of 3 larvae per condition. After 24 hours, larval hemolymph was removed directly in anticoagulation buffer, centrifuged at 10,000 *xg* for 5 minutes, and resuspended in coagulation buffer.

Brightfield microscopy images were randomly taken at 4× magnification, with 15 to 20 images taken per condition per replicate. These images were analyzed using Fiji [134] Particle Analyzer function using with a threshold set between 0 and 120 mean gray value. Particle area and numbers were calculated. Additional images were taken of nodules at 20× and 100× magnification, the latter of which were used to manually score the degree of melanization of fungal

cells within the nodules. Statistical significance of differences between melanized particle area was analyzed using a nested nonparametric Mann–Whitney–Wilcoxon rank test using the *nestedRanksTest* package (*Version 0.2*, D.G. Scofield, 2014) [77] in R for R 4.0.2 GUI 1.72 for Mac OS at https://www.r-project.org/ (R Core Team, 2020).

### *Anopheles gambiae* phenoloxidase activity

Phenoloxidase activity assays were performed as previously described [132]. Experiments were done in biological triplicate with different batches of mosquitoes, as well as in technical triplicate per biological replicate of 3 batches of 10 mosquitoes.

### *Anopheles gambiae* survival

Adult female mosquitoes of *A*. *gambiae* Keele strain were raised on 10% sucrose for 3 days postemergence. On the third day, adult females were sorted into 7 groups of 40 and placed into mesh-covered cardboard cups and provided a cotton ball with 10% sucrose mixture with either 0 μM (Control), 30 μM, 100 μM, 300 μM, 1 mM, 3 mM, or 10 mM glyphosate. The cotton balls were replaced every third day with new cotton balls and fresh sucrose/glyphosate solutions. Mosquito death was monitored daily for 14 days. Experiments were performed in 3 independent replicates, for a total of 120 mosquitoes in each treatment group.

### *Anopheles gambiae* cuticle pigmentation and wing size

Adult female mosquitoes were drugged for 5 days as previously described. Mosquitoes were cold euthanized and mounted dorsally on a slide with double-sided tape. Images of the mosquito ventral abdomen were taken under a dissection microscope with constant exposure and lighting conditions. Pigmentation was measured using Fiji software [134]. The entire abdomen of each mosquito was selected using a freehand selection tool, and the 8-bit mean gray value was measured using the Measure tool. A measurement of 0 corresponds to a pure black gray value, whereas 255 corresponds to a pure white gray value.

Following abdomen pigmentation measurements, mosquito bodies were removed, with careful attention to keeping the wings remaining intact on the tape. Intact wings were imaged on a microscope, and the length of the individual wing lengths were measured from tip to tip using Fiji Measure tool.

### *Anopheles gambiae* infection with *Plasmodium falciparum*

Adult female mosquitoes (3 to 4 days old) of *A*. *gambiae* Keele strain were sorted and drugged as described above. On the fifth day of glyphosate exposure, mosquitoes were provided a blood meal containing *P. falciparum*. Blood-fed engorged mosquitoes were sorted on ice and fed 10% sucrose ad libitum for 8 days. Midguts were dissected and stained with 0.2% Mercurochrome solution, and oocysts were enumerated using a 20X objective with light microscopy. See S9B Fig.

### *Anopheles gambiae* midgut microbiome analysis

Adult female mosquitoes (3 to 4 days old) of *A*. *gambiae* Keele strain were sorted and drugged as described above. On the fifth day of glyphosate exposure, mosquitoes were sterilized in ethanol for 2 minutes, washed, and dissected in sterile PBS. The midguts were removed, placed in 500 μl sterile PBS on ice, homogenized, diluted, and plated on LB agar plates. Plates were incubated at 30°C for 3 days, and individual colonies were counted. Each experiment used 10 to 20 mosquitoes per condition, and the experiment was performed 3 independent times.

For the 16S rRNA sequencing studies, mosquitoes were reared and drugged, and then midguts were dissected as described above, with 5 individual midguts per condition. DNA was extracted from frozen mosquito samples using the Lucigen EpiCentre MasterPure DNA extraction kit (Lucigen, Middleton, Wisconsin, USA). The bacterial 16S rRNA gene was amplified by PCR, and sample-specific Illumina adapters were ligated to the PCR products. PCR products from multiple samples were pooled and sequenced on the Illumina MiSeq platform by the University of Connecticut MARS Facility. Data were then analyzed using mothur [135] to construct contigs to align forward and reverse reads, remove ambiguous bases and chimeric regions, align sequences to the Silva 16S V4 reference database, and cluster reads into 3% operational taxonomic units (OTUs). Sequences derived from known contaminants were selectively removed. Alpha and beta diversity measurements were performed using the Shannon diversity index and Bray–Curtis dissimilarity distance, respectively. Bray–Curtis distances were graphed on principal coordinates analysis (PCoA) plots in 2 dimensions. Taxa and PCoA graphs were produced using MicrobiomeAnalyst [136,137]. See S9C Fig.

### Dopaquinone formation MBTH assay

Quinones like dopaquinone are unstable and difficult to study directly; thus, dopaquinone quantification relies on the formation of a stable adduct with MBTH (3-methyl-2-benxothiaxolinone hydrazine) that forms a pigment that absorbs at 505 nm [138]. This absorption overlaps with the absorption of another melanin intermediate, dopachrome (Q), but is not expected to interfere since dopaquinone reaction with MBTH prevents dopachrome formation. Further, the molar absorbance coefficient for MBTH-Dopaquinone is more than 10 times higher (39,000 L/[mol cm]) than that of dopachrome (3,700 L/[mol cm]), and interference from dopachrome would be relatively small.

MBTH reaction mixtures were prepared as previously described [138]. This mixture is warmed at 42˚C to help solubilize the components. Then, 5 μl of 2 μg/ml mushroom tyrosinase (Sigma, Product #T382) and 20 μl of 20 mM L-DOPA are added to the MBTH solution, and 160 μl of the solution is immediately added to each well containing compounds. The plate was read at an absorbance of 505 nm for 30 minutes at 30˚C and read again at 1 hour and overnight. The dopaquinone levels are determined by the formation of the bright pink adduct between the quinone and the MBTH.

### Dopachrome and melanin measurements

Tyrosinase activity was determined as previously described [132], substituting mushroom tyrosinase for phenoloxidase. The formation rate of dopachrome is measured as the maximum velocity of this reaction, and the dopachrome levels are measured as the absorbance at 490 nm after 30 minutes as the absorbance values plateau. Melanin levels are measured as the absorbance at 490 nm after the reaction has continued for 5 days in the dark at room temperature.

### Free-radical scavenging ABTS assay

ABTS solution was prepared as previously described [139]. To test the radical-scavenging capability of the compounds, 10 μl of the compounds were serially diluted in a 96-well plate as previously described, and 90 μl of diluted ABTS was added to each well. The 734 nm absorbance was measured immediately, after 10 minutes, 1, and 2 hours. In kinetics experiments, absorbance readings were taken every 2 minutes for 5 hours.

To measure the radical scavenging capacity of the synergistic compounds and L-DOPA mixtures, ABTS was prepared and diluted in Milli-Q water. In each well, 5 μl of compound stocks were added with either 5 μl of water or 5 μl of 500 μM L-DOPA. Next, 90 μl of ABTS

solution was added to the well, and the absorbance was read immediately at 734 nm. Synergy was calculated from these data using the following formula:

$$\text{Synergy Ratio} = \frac{(\Delta\text{Abs 734 Compound Alone} + \Delta\text{Abs 734 DOPA Alone})}{\Delta\text{Abs 734 Compound with DOPA}}$$

## Glyphosate effect on L-DOPA

To determine if L-DOPA is reacting with glyphosate, we analyzed by NMR. We diluted 300 mM stock of glyphosate in water to 60 mM (10 mg/ml) in $D_2O$, prepared 20 mM (4 mg/ml) L-DOPA in $D_2O$, and prepared 2 mixtures of glyphosate and L-DOPA: one with 20 mM (4 mg/ml) L-DOPA and 60 mM (10 mg/ml) of glyphosate in $D_2O$, and another with a low concentration of 1 mg/ml for both compounds equaling 5 mM L-DOPA and 6 mM glyphosate. We then performed $^{31}P$-NMR and $^{1}H$-NMR on these samples.

## Glyphosate effect on tyrosinase

To determine the tyrosinase kinetics with glyphosate as an inhibitor, we serially diluted 155 μl of 20 mM L-DOPA in Milli-Q water. To each dilution of L-DOPA, we added 20 μl of glyphosate diluted in PBS and 5 μl of 2 μg/ml mushroom tyrosinase to the reaction mix. In order to account for nonenzymatic oxidation of L-DOPA, we ran an experiment in parallel, in which we added 5 μl of Milli-Q water instead of tyrosinase. The reaction mix was kept at 30°C for 24 hours. The plate was read at 490 nm. To calculate enzyme-specific oxidation of L-DOPA, the no enzyme values were subtracted from the tyrosinase rows. The kinetics curve is plotted as a function of absorbance after 24 hours of reaction time versus concentration of L-DOPA.

We tested if tyrosinase concentration had an effect on the percent inhibition of the reaction. We prepared dilutions of tyrosinase. We added 5 μl of each dilution to a 96-well plate and added 135 μl of Milli-Q water, 20 μl of 20 mM L-DOPA, and 20 μl of glyphosate in PBS. We measured maximum velocity of this reaction at 490 nm. The difference in velocities and percent inhibition reported were calculated by difference = $V_{max\ water} - V_{max\ glyph}$, and percent inhibition = $100^{*}(V_{max\ glyph}/V_{max\ water})$.

To determine if glyphosate irreversibly affects tyrosinase activity, 450 μl of 20 μg/ml mushroom tyrosinase was prepared in 450 μl of 50 mM sodium phosphate buffer (pH 7), either with 50 μl of 300 mM glyphosate or 50 μl of Milli-Q water. The enzyme solution was loaded into a hydrated 10,000 MWCO Slide-a-lyzer dialysis cassette (Thermo Scientific, Waltham, Massachusetts, USA), and the enzyme solutions were dialyzed in a 50-mM sodium phosphate buffer at 4°C, according to the manufacturer's protocol. Protein concentrations were measured and normalized using sodium phosphate buffer. To measure the kinetics of the control enzyme versus the treated enzyme, a kinetics assay was performed as previously described. Each reaction's maximum velocity is determined and plotted.

## Copper rescue of melanin inhibition

As previously described, serial dilutions of glyphosate were arrayed in 8 rows; 1 row per copper ion concentrations to be tested. Copper sulfate was prepared and serially diluted, and 10 μL of the copper solution is added to each well containing the glyphosate dilution. To each well, 150 μL of reaction mix (125 μL of Milli-Q water, 20 μL of 20 mM L-DOPA, and 5 μL of 2 μg/mL mushroom tyrosinase (5 μl of water used for auto-oxidation experiments) was added. The final copper ion concentrations were 400, 200, 100, 50, 25, 12.5, 6.25, and 0 μM. The dopachrome and melanin measurements are reported as previously described.

## Cyclic voltammetry

Cyclic voltammetry was performed using a Metrohm Autolab potentiostat (Switzerland), 3 mm Glassy Carbon working electrode, 10 mm × 10 mm × 0.1 mm platinum plate counter electrode, and an Ag/AgCl reference electrode in 3 M KCl solution. Solutions were prepared in 0.1x PBS (Difco, Franklin Lakes, New Jersey, USA) at a pH 6.00, adjusted with NaOH and HCl. A volume of 10 mL of L-DOPA solution was freshly prepared in this buffer, and 1 mL of glyphosate, glycine, water, etc., solution at pH 6.00 were added to the L-DOPA. Readings were done with 3 tracings at a scan rate of 50 mV/s at intervals of 5 mV steps. Glassy carbon electrode was washed and polished between readings with slurry of alumina powder and water on cloth pads.

## Supporting information

**S1 Fig. *G. mellonella* supplemental data. (A)** Broad-spectrum protease inhibitor (cOmplete, Roche, Basel, Switzerland) was added to *G. mellonella* hemolymph to prevent the activation of new phenoloxidase and to control for any impact that glyphosate may have on phenoloxidase activation cascade, cell viability, and gene expression. The general trend remains the same that glyphosate inhibits phenoloxidase activity with and without protease inhibitor, albeit lower with protease inhibitor due to the lower concentration of activated enzyme. **(B)** Phenoloxidase activity was assessed using exogenous L-DOPA for 1 batch of *G. mellonella*, during these experiments, the lower concentration of glyphosate resulted in increased phenoloxidase activity as compared to the control. This suggests that there may be some cellular regulation of phenoloxidase induced by glyphosate. It is possible that the doses of glyphosate tested elicit some cellular response that increases phenoloxidase expression, secretion, and/or activation as a feedback/hormesis-like response to the reduced melanin production. These data represent 3 independent replicates, but this pattern of enzymatic activity as a function of glyphosate concentration was not seen in subsequent batches of larvae. **(C)** Hemocyte viability was not dramatically affected by concentrations of glyphosate ranging from 100 μM to 10 mM, indicating that our data are likely not artifacts of cytotoxic concentrations of glyphosate. Error bars in **(A–C)** represent ±SD. **(D)** AMPA, a major metabolite of glyphosate, inhibits tyrosinase-mediated melanization similar to glyphosate. Grayscale bars represent mean absorbance at 490 nm relative to no compound control. The darker colors correspond to increased pigment formation. **(E)** Larvae treated with glyphosate and subsequently infected with *lac1Δ* mutant *C. neoformans* strain showed a similar pattern of increased susceptibility as the wild-type H99, although the differences in susceptibility with the *lac1Δ* infected larvae are not statistically significant. Each experiment represents at least 3 independent replicates. The PBS mock infection condition represents survival of 95 animals, over the span of 4 biological replicates, and 6 total technical replicates. The *lac1Δ* mutant infection represents survival of 75 animals over the span of 4 biological replicates. The PBS mock infection data are the same as the data in Fig 5B, as all the infections were done concurrently under the same conditions. **(F)** Single injection of 10 μl of 1 mM glyphosate does not affect the pupation of *G. mellonella* at 30˚C and RT. Data from 30˚C represent 25–35 animals for each group over 2 biological replicates, and data from RT represent 45 animals from each group over 3 biological replicates. Statistical analysis performed using log-rank Mantel-Cox tests. **(G–I)** The 3 individual replicates from **Fig 1D** showing the size of the dark melanized particles within nodules are significantly smaller in the glyphosate-treated infected groups compared to the PBS-treated infected groups, with **(G)** and **(H)** showing that there were more melanized spots in the PBS-treated infected group compared to the glyphosate-treated. All statistical analyses performed using GraphPad Prism version 8.4.3 for Mac OS, GraphPad Software, San Diego, California, USA, www.graphpad.com.

For underlying data, please see Data Availability section and/or S1 Table. AMPA, amino-methylphosphonic acid; Glyph, glyphosate; L-DOPA, 3,4-dihydroxyphenylalanine; RT, room temperature.
(TIFF)

**S2 Fig. Low efficiency Plasmodium *falciparum infection* of *A. gambiae* and effects on mosquito cuticle. (A)**. Oocyst count per midgut for mosquitoes treated with or without glyphosate and infected with high-passage *P. falciparum* gametocyte culture, resulting in a low efficiency infection. Data represent 1 biological replicate. Dotted black line indicates y = 0. Black lines for each condition indicate median oocyst count per midgut. We have chosen not to include the data from this replicate in the data shown in Fig 6, because the results from this one-off replicate appear due to poorly infectious parasite culture. Additionally, it is difficult to make comparisons using the low infection burden of the control group with the treatment groups, as well other replicates with higher oocyst burdens. **(B)** Infection prevalence (percent midguts with at least 1 oocyst) from the experiment described in **(A)**. Fisher exact test performed for each condition individually compared to control and corrected for multiple comparisons using the Bonferroni method. **(C)** A total of 5 days of 1 mM glyphosate treatment in adult female mosquitoes does not influence the abdomen's cuticular darkness as measured by mean gray value with 0 being pure black and 255 being pure white. Data representative of 2 biological replicates with 88 mosquitoes measured per condition. **(D)** Wing length, as a proxy for body mass and size, is not affected by 5 days of glyphosate treatment. Data representative of a single biological replicates with 32–36 mosquitoes measured per condition. Line and error bar represent mean ± SD in **(C, D)**. Unpaired *t* test performed to determine statistical significance in **(C, D)**. All statistical analyses performed using GraphPad Prism version 8.4.3 for Mac OS, GraphPad Software, San Diego, California, USA, www.graphpad.com. For underlying data, please see Data Availability section and/or S1 Table. Glyph, glyphosate; n.s., not significant.
(TIFF)

**S3 Fig. Glyphosate affects the *A. gambiae* microbiota in a dose-independent manner. (A)** At the class level, glyphosate leads to an enrichment of Alphaproteobacteria and a depletion in Gammaproteobacteria. Tables showing the relative abundance of bacterial classes **(B)** and individual bacterial genera **(C)** following glyphosate treatment. **(D)** Alpha diversity does not follow a distinctive pattern with increasing glyphosate dose. **(E)** Glyphosate-treated and control-treated microbiota cluster separately in ordination space, but the clusters are not dose dependent. For underlying data, please see Data Availability section and/or S1 Table.
(TIFF)

**S4 Fig. Reaction of glyphosate with L-DOPA.** Representative $^1$H NMR spectra of 60 mM glyphosate solution in $D_2O$ (**Green**), 20 mM L-DOPA solution in $D_2O$ (**Red**), and 20 mM L-DOPA mixed with 60 mM glyphosate in $D_2O$ (**Blue**). There appears to be no shift in $^1$H peaks and no appearance of new peaks, which is indicative of no reaction occurring between the compounds. Data representative of findings from 3 independent replicates. For underlying data, please see Data Availability section and/or S1 Table.
(TIFF)

**S5 Fig. Glyphosate appears to "buffer" copper concentration in solution.** High doses (2–16 mM) of glyphosate prevent the enzymatic activity enhancing effects of lower copper concentration (6.25–25 μM), but high doses of glyphosate also prevent the enzyme inhibitory effects of high copper concentration (100–400 μM). Error bars represent ±SD. Data represent 2 independent replicates. For underlying data, please see Data Availability section and/or S1 Table.
(TIFF)

**S6 Fig. Antioxidant properties of glyphosate. (A)** Change in absorbance of ABTS solution at 734 nm over time for 33.33 mM glyphosate relative to the no glyphosate control. This indicates glyphosate quenches free radicals over an extended period of time. **(B)** Calculated antioxidant radical scavenging synergy between compounds tested and L-DOPA. Values represent the mean of at least 3 independent replicates. Error bars represent ±SD. For underlying data, please see Data Availability section and/or S1 Table. ABTS, 2,2′-azino-bis(3-ethylbenzothiazo-line-6-sulfonic acid); L-DOPA, 3,4-dihydroxyphenylalanine.
(TIFF)

**S7 Fig. Glyphosate inhibits melanin production independent of L-DOPA concentration.** (**A**) Inhibitory concentrations of glyphosate are not affected by L-DOPA concentration. This indicates that glyphosate is not reacting proportionally with L-DOPA as measured by absorbance at 490 nm after 5 days of reaction, relative to the no glyphosate control and with background absorbance subtracted. (**B**) The $IC_{50}$ of glyphosate remains constant at approximately 1 mM relative inhibition of melanin production appears dependent on glyphosate concentration alone and not on L-DOPA to glyphosate ratio. Data in panels (**A**) and (**B**) represent 2 alternative visualizations of the same experimental data. Error bars represent ±SD. Each experiment represents at least 3 independent replicates. Grayscale bars represent mean absorbance at 490 nm relative to no compound control. The darker colors correspond to increased pigment formation. Red line represents the approximate $IC_{50}$. Crossed out boxes represent values with no data. For underlying data, please see Data Availability section and/or S1 Table. L-DOPA, 3,4-dihydroxyphenylalanine.
(TIFF)

**S8 Fig. Cyclic voltammetry supplemental data. (A)** Peak 1 was validated as the oxidation of L-DOPA, and Peak 2 was validated as the reduction peak of dopaquinone by correlating increased peak intensity with increasing concentration of L-DOPA under the same potentiostat parameters. **(B)** Glycine (16 mM)—a non-phosphate analog of glyphosate, a noninhibitor of melanization, and a nonantioxidant—does not alter the oxidation potential of L-DOPA. Conversely, citric acid (16 mM)—a known synergistic antioxidant and inhibitor of melanization—does alter the oxidation potential of L-DOPA in similar ways as glyphosate. The L-DOPA alone control and glyphosate voltammograms in panel (**B**) are the same as those found in **Fig 7E**. Each experiment represents at least 3 independent replicates, with 3 cycles per replicate. The tracings represent the mean value of the 3 replicates over the course of 3 cycles. For underlying data, please see Data Availability section and/or S1 Table. L-DOPA, 3,4-dihydroxyphenylalanine.
(TIFF)

**S9 Fig. Experimental methods diagram for insect experiments. (A).** During *G. mellonella* infection with *C. neoformans*, larvae were injected with 10 μl of 1 mM glyphosate, left to recover for 5 hours, and were subsequently infected with $10^4$ cells/larvae of *C. neoformans* H99 strain. Survival was monitored for 14 days. **(B)** During *A. gambiae* infection with *P. falciparum*, mosquitoes were drugged with glyphosate-laced 10% sucrose solution for 5 days, then fed with a *P. falciparum*-infected blood meal, and fed 10% sucrose for 8 days. On day 8, mosquitoes were dissected, and the midguts were stained with mercurochrome to facilitate oocyst enumeration. **(C)** Glyphosate-drugged mosquitoes were dissected under sterile conditions, and 5 midguts were collected individually per condition. DNA was extracted from samples and bacterial 16S rRNA genes were amplified by PCR and sample-specific Illumina adapters were ligated to products. PCR products were pooled and sequenced on the Illumina MiSeq

platform. Data were then analyzed using mothur to construct contigs, align reads, remove ambiguous bases and chimeric regions, align sequences to the Silva 16S V4 reference database, and cluster reads into 3% OTUs) Sequences from known contaminants were removed. Alpha and beta diversity measurements were performed using the Shannon diversity index and Bray–Curtis dissimilarity distance, respectively, and plotted using MicrobiomeAnalyst. Figures made with BioRender. OTU, operational taxonomic unit.
(TIFF)

**S1 Table. Links to the underlying data for each figure in the Figshare data repository.**
(XLSX)

## Acknowledgments

We would like to acknowledge Dr. Gene Fridman in the Johns Hopkins School of Medicine for lending the use of the Fridman lab's Metrohm Autolab potentiostat for use in the cyclic voltammetry experiments. We would like to thank the Johns Hopkins Malaria Research Institute and the Department of Molecular Microbiology and Immunology Insectary and Parasite Core facility, and Dr. Joel Tang from the Johns Hopkins NMR Core facility for his help with the NMR experiments. We would also like to thank the entire Casadevall Lab for their suggestions and inputs during conversations and their feedback during lab meetings and presentations. Figures in S9 Fig were made using BioRender.

## Author Contributions

**Conceptualization:** Daniel F. Q. Smith, Emma Camacho, Raviraj Thakur, Alexander J. Barron, Nichole A. Broderick, Arturo Casadevall.

**Data curation:** Daniel F. Q. Smith, Alexander J. Barron.

**Formal analysis:** Daniel F. Q. Smith, Alexander J. Barron.

**Funding acquisition:** Emma Camacho, George Dimopoulos, Nichole A. Broderick, Arturo Casadevall.

**Investigation:** Daniel F. Q. Smith, Emma Camacho, Alexander J. Barron, Nichole A. Broderick.

**Methodology:** Daniel F. Q. Smith, Emma Camacho, Raviraj Thakur, Alexander J. Barron, Yuemei Dong, Nichole A. Broderick, Arturo Casadevall.

**Project administration:** Daniel F. Q. Smith, Emma Camacho, Alexander J. Barron, Nichole A. Broderick, Arturo Casadevall.

**Resources:** Emma Camacho, Raviraj Thakur, George Dimopoulos, Nichole A. Broderick, Arturo Casadevall.

**Software:** Alexander J. Barron.

**Supervision:** Emma Camacho, Raviraj Thakur, Nichole A. Broderick, Arturo Casadevall.

**Validation:** Daniel F. Q. Smith.

**Visualization:** Daniel F. Q. Smith, Alexander J. Barron.

**Writing – original draft:** Daniel F. Q. Smith.

**Writing – review & editing:** Daniel F. Q. Smith, Emma Camacho, Raviraj Thakur, Alexander J. Barron, Yuemei Dong, George Dimopoulos, Nichole A. Broderick, Arturo Casadevall.

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
