## [Editor Report · Decision Letter 0]

17 Jun 2020

Dear Dr Casadevall,

Thank you very much for submitting your manuscript entitled "Glyphosate Inhibits Melanization and Increases Insect Susceptibility to Infection"as a Research Article for review by PLOS Biology. Thank you also for your patience as we completed our editorial process, and please accept my apologies for the delay in providing you with our decision.

We do appreciate the significance of your study analysing the mechanisms underlying melanin inhibition by the herbicide glyphosate (GLYPH) both in vivo and in vitro, and your results showing that GLYPH inhibits dopaquinone production in a dose-dependent manner, and that this disrupts the production of dopachrome and melanin, and the general oxidative properties of melanogenesis in an in vitro model. In addition, you show that GLYPH impedes insects melanogenesis by inhibiting phenol oxidases activation in a dose-dependent manner and that it increases susceptibility to infection. While your findings will be certainly interesting for other researchers in this field and we do appreciate that you identify specific steps in melanogenesis affected by this herbicide, we do think that the general conceptual advance is incremental to what was already known in the literature. Thus, I am sorry to say we are not persuaded that the novel insights provide the strength of advance that we are striving to achieve for PLOS Biology.

I am sorry that we cannot be more positive on this occasion. I hope you appreciate the reasons for this decision and will consider PLOS Biology for other submissions in the future. Thank you for your support of PLOS and of Open Access publishing.

Sincerely,

Ines

--

Ines Alvarez-Garcia, PhD

Senior Editor

PLOS Biology

Carlyle House, Carlyle Road

Cambridge, CB4 3DN

+44 1223–442810

---

## [Decision Letter · Decision Letter 1]

7 Sep 2020

Dear Dr Casadevall,

Thank you very much for submitting your manuscript "Glyphosate Inhibits Melanization and Increases Insect Susceptibility to Infection" for consideration as a Research Article at PLOS Biology. Thank you also for your patience as we completed our editorial process, and please accept my apologies for the delay in providing you with our decision. Your manuscript has been evaluated by the PLOS Biology editors, an Academic Editor with relevant expertise, and by three independent reviewers.

The reviews of your manuscript are appended below. You will see that the reviewers find the work potentially interesting. However, both reviewers 1 and 3 have several suggestions to improve the manuscript by providing clarifications, a better background introduction that includes missing references and improving the writing for a broad audience. Reviewer 2 is more critical and thinks that immune melanogenesis could be a specialised pathogen/wound response in relatively few insects, and that GLYPH may similarly affect laccases and phenol oxidases/tyrosinases produced in other tissues that perhaps play even more important roles in insect development and reproduction, and the effects observed might be due to indirect causes.

Based on the specific comments and following discussion with the academic editor, I regret that we cannot accept the current version of the manuscript for publication. We remain interested in your study and we would be willing to consider resubmission of a comprehensively revised version that thoroughly addresses all the reviewers' comments, paying particular attention to those from Reviewers 2 and 3. We cannot make any decision about publication until we have seen the revised manuscript and your response to the reviewers' comments. Your revised manuscript would be sent for further evaluation by the reviewers.

We appreciate that these requests represent a great deal of extra work, and we are willing to relax our standard revision time to allow you six months to revise your manuscript.We expect to receive your revised manuscript within 6 months.

**IMPORTANT - SUBMITTING YOUR REVISION**

*Resubmission Checklist*

*Published Peer Review*

*PLOS Data Policy*

*Blot and Gel Data Policy*

Sincerely,

Ines

--

Ines Alvarez-Garcia, PhD,

Senior Editor,

ialvarez-garcia@plos.org,

PLOS Biology

Reviewers’ comments

Rev. 1:

This is well-designed and timely study that extends previous findings showing that glyphosate inhibits melanization in fungi. In this study, Smith et al. demonstrate the molecular mechanisms by which glyph inhibits melanization and extend the effects to insects: they show that glyph also inhibits melanization in wax moth larvae and in malaria-transmitting mosquitoes, which leads to increased susceptibility to infection. I only have minor concerns regarding the study, which are described below.

Abstract, lines 30-32: It seems this sentence is missing a link after the word melanization, such as an "and".

Lines 190, 507: Have you considered that glyph could be reversibly inhibiting tyrosinase, such as it does for the EPSP synthase, the main known target of glyph in plants and microbes? In this case, removing glyphosate or increasing the concentration of substrates would dislodge glyphosate from the enzyme, which would become active again.

Line 240: Figure citation is wrong

Line 346: Figure 5a-b is not cited

Figure 6b: are there statistically significant differences in percent survival between groups? In other words, did you perform statistical analyses to corroborate what is said in lines 421-424?

Figure 6c,d: it is not clear if statistical corrections were performed after multiple comparisons

Lines 453: The use of CFU counts to investigate microbiome density/load is not appropriate, since not all microbes will grow under specific lab conditions. Ideally, quantitative PCR should be used to assess microbial loads

Lines 459, figure 7c,d: I could not find statistical support for the observed changes in alpha and beta diversity measures. Without statistically significant support, you cannot rule out that glyph affects A. gambiae microbiome in a dose-independent manner

Why are the glyph-treated groups analyzed together in figure 7c,d, and not individually as in supplementary figure 8?

Rev. 2:

This manuscript describes two very different types of experimental studies, but the authors fail to provide a substantive basis for why they are presented together. For these reasons, I found the manuscript poorly organized and hard to follow, and rife with biased speculation about the consequences of GLYPH use.

First, in vitro studies demonstrated how glyphosate (GLYPH), a widely used herbicide, affects the melanization reaction of a fungal tyrosinase. The enzyme was incubated with GLYPH at different doses and conditions in the presence of different substrates and inhibitors. This work appears to be the first to investigate GLYPH inhibition of tyrosinase activity as no references for related studies were offered, so it could stand on its own in a separate manuscript submitted to a specialized journal.

The next set of studies examined the effects GLYPH on different aspects of melanogenesis in wax moth larvae and mosquito females. The authors appear to have little knowledge of related physiological processes in insects, fail to provide key background information about the insect models, and over interpret the importance of their results. The continued use and efficacy of GLYPH is highly controversial, and the authors should offer an unbiased over-view of the literature that encompasses the importance of GLYPH for inexpensive weed management to enable global food production along with possible environmental effects, including the so called 'insect apocalypse' that is only anecdotally reported for a few "instagram-able" insect groups and life stages, and worse still the bases for reports relying on meta-analyses,. Termites and ants largely inhabit soils and make up an estimated quarter to a third of the planet's animal mass - where's the data showing their decline or that of cockroaches or mosquitoes?

The author's premise for this work is that melanogenesis, one of several components of the insect immune response, may be affected by GLYPH persistence in the environment. No general description/comparison of cellular and humoral defenses, which would include melanogenesis, in insects was provided by the authors. My sense is that immune melanogenesis is a specialized pathogen/wound response in relatively few insects (e.g. melanization of parasitoid eggs in lepidopteran larvae), which the authors do not cover, and more properly, the emphasis should be effects of GLYPH on the phenol oxidase cascade as part of humoral immunity. The authors chose two very different insects and life stages for their studies, but offer no up-to-date reviews or references about this process specific to the insects (or related species).

Furthermore, the authors do not mention that given the above enzymatic characterization, GLYPH may similarly affect laccases and phenol oxidases/tyrosinases produced in other tissues that play perhaps even more important roles in insect development and reproduction. Such enzymes in the insect epidermis facilitate sclerotization and hardening of new cuticle after molting, and in the nervous system, dopamine produced by tyrosinase is an important neurotransmitter. Thus, the effects of GLYPH on wax moth survival after fungal infection primarily may be due to the failure to molt (altered dopamine neurotransmission?) or harden cuticle, given that larvae in the study presumably were fed and should have gone through one or molts within the 14 days they were monitored (Fig. 5 B). No mention of molting was made, and no direct evidence for melanization of the fungal pathogen in the larvae was given for the controls, so what link between GLYPH inhibition of melanogenesis and the multi-armed immunity in this insect is demonstrated by the data? None - just speculation - a more focused effort to assess the effects of GLYPH on other key immune pathways could possibly elevate the significance of this work. As for the mosquitoes, perhaps GLYPH doses varied in their interference with neural transmission thus affecting survival after treatment (Fig. 6B). If there is no melanization of infective Plasmodium oocysts in GLYPH treated mosquitoes, then what is the link between GLYPH treatment, melanogenesis, and immunity that has a direct effect on Plasmodium infection and prevalence (Fig. 6D & D)? The fact that GLYPH is known to slow the growth of Plasmodium in cell culture may alone explain these results (Phillips, H. Could malaria be killed by a garden weedkiller? Nature (1998). https://doi.org/10.1038/news980702-2) - a point not addressed by the authors. The inconsistent effects of GLYPH doses on the mosquito microbiota may also be due to direct inhibition of microbial pathways, as suggested by the authors, but that did not limit the speculation about melanogenesis, immunity, and the gut microbiota. Overall, the effects of GLYPH treatment on unrelated physiological processes in the two insects had little significance or coherence.

Specific concerns:

Figures are not covered in the Results in an orderly manner (e.g. line 410, Fig. 6C is presented before 6B, and 6A is mentioned in the wax moth results section but not in the mosquito section). Several figure panels are not adequately described in the captions (e.g. Fig. 1A, and Fig. 2E what is shown in the inset?) and what exactly do the panels with black, gray, and white rectangles represent in Fig. 2D & E, Fig. 3D, Fig. 5C, and Sup. Fig. 4B, and where is the data analysis?

In the caption for Fig. 6B, it is stated that "survival curves represent 120 animals across three biological replicates", so by my calculation that would be 12 mosquitoes for each of the 10 treatments, which for most survival studies is way too few. The numbers of Plasmodium oocysts per midgut given for treated Anopheles females are exceptionally high in Fig. 6C compared to most studies and even in Sup. Figure 7, which is more commonly reported, but no explanation is offered for the count differences based on methodology or treatment between the mosquito cohorts or data sets.

Rev. 3: Sharon Pochron – this reviewer has waived anonymity

Review of: Glyphosate Inhibits Melanization and Increases Insect Susceptibility to Infection (Glyphosate Inhibits Melanization in Insects and Fungi).

Overview: In a super interesting paper, the authors run a number of experiments that show that glyphosate exposure reduced wax moth larvae survival after infection, increased parasite burden in malaria-transmitting mosquitoes, and altered midgut microbiome composition in adult mosquitoes. The authors drew a line from their findings to the insect apocalypse, which makes the paper very powerful indeed.

Weaknesses: The manuscript is filled with grammatical errors, including run-on sentences. It is also poorly referenced, leaving out many important papers in the field, and using questionable references (e.g. meeting abstracts) for critical points, which is particularly annoying since solid references exist. I highly recommend the authors spend a day using Web of Science. Additionally, many of the key sections of the paper, even the title, are poorly argued.

Strengths: The experiments, the results, and the implications stemming from the results are super important. The figures and illustrations are killer.

Recommendation: Accept with major revisions, especially of the title, abstract, Introduction and Discussion.

Title: The long title is acceptable, but the short title makes no sense. The short title implies that the authors ran experiments on fungi, which they didn’t. The abstract doesn’t discuss fungi. Readers of the short title are left thinking, “Wait, what? Where’s the fungi?”

Abstract: Line 29, I’m not sure I’d use “environmental conditions.” I think you really mean something like “exposure to ubiquitous contaminants.”

Lines 30-32: The authors state, “Here we elucidate the mechanism underlying glyphosate’s inhibition of melanization demonstrate the herbicide’s multifactorial effects on insects.” The sentence makes no sense.

Lines 30-33 should provide the set-up for the findings, which start on line 33. But I can’t tell what the set-up is. The abstract should tell me why they ran the experiments that they ran. It doesn’t.

Line 33: While technically, glyphosate is a drug, I would use a more specific word here. Contaminant? Herbicide?

Line 36: so-called gets a hyphen.

Introduction: So many run-on sentences. The first comma on line 46 should be a period, for example. Same for the comma on line 52. Regarding the references in the first paragraph (lines 44 – 53), 3 of the 4 references cited are about insects—if you want to say something about ubiquity of tyrosinases, I suggest that you reference at least one more general paper about it. Also, if you care enough about tyrosinases to put it into you keywords, you might want to state that they it is amongst the most widely employed enzymes as “green catalysts” for environmental applicability—and include a reference to that affect.

Line 61, the comma should be a semicolon.

Line 63: You can’t use a mouse paper to reference the statement “Glyphosate is a widespread herbicide found in the environment.” There are a gazillion papers out there that show that glyphosate is everywhere. Start with Battaglin et al., 2014 and include Myers et al., 2016. Consider Bach et al., 2018 and Sihtmae at al., 2013. Laitinen et al. have a suite of papers on the topic. Additionally, glyphosate itself isn’t the herbicide. It is the active ingredient in a family of herbicides. No one applies glyphosate alone to kill weeds. You make this point effectively in the next paragraph, so maybe consider revising your paragraph ordering?

Still focusing on the paragraph that starts on line 63, I’m not sure what it’s function is. What are you trying to communicate to the readers with it? I think you’re trying to make the point that glyphosate inhibits melanization in fungus, but you start with the widespread contamination point and end with cosmetics. You need to revise your focus here.

Regarding your paragraph starting on line 71, you need to ask yourself what point you’re trying to make here. It’s highly unfocused. You start with the mechanism, wander into global contamination (again, and while missing major references on the topic; see above), wander in human cancers, and end on need to understand environmental impacts. That is a lot to ask of one paragraph.

Regarding the paragraph that starts on line 84, you are missing critical references. It should include Battaglin et al., 2014 and Gill et al. 2018. Gill et al., 2018, reviews the impact of glyphosate on all animals. Your review of the impact of glyphosate on other insects is depauperate. You should include: (Baglan et al., 2018) (mosquitos); (Tahir et al., 2019) (spiders); (Farina et al., 2019; Tomé et al., 2020) (bees).

Also, like your earlier paragraphs, the paragraph that starts on line 84 is unfocused, in addition to being poorly referenced. If you’re going to discuss the half life of of glyphosate, you should be using more recent review papers such as (Singh et al. 2020) and (Van Bruggen et al. 2018). If you’re going to discuss how long it can remain in the top layer of soil, you should be citing all of the Laitinen papers. There is a rich body of literature discussing the impact of glyphosate on soil microbes and fungi. Look at Pochron et al. 2020 for a review of that.

But again, pick the point you want to make and make it in one paragraph. This paragraph wanders from run-off, to environmental concentrations, to the impact on soil microbes and its subsequent impact on plants, back to environmental concentrations, to the impact on algae, to the impact on bee microbiomes and fly microbiomes. No reader can make sense of the story you’re trying to tell.

In your final Introduction paragraph (line 96), you introduce the reader to a mushroom tyrosinase model with no explanation. You toss in a bunch of scientific names, without the common names, and you sketch out your experiment. At this point in the Intro, I should be able to determine how and why your experiment is important. It should tell me why you, the authors, think that glyphosate impacts melanin production. You don’t. I have no idea why you’re running these experiments.

Lastly, your Introduction and Discussion should bear some resemblance to each other. Your Discussion opens with AMPA, but AMPA is not discussed in your Intro.

For each of the outcome variables you measure in your Results (and you have many! All interesting!), you need to touch on those in your Intro. What impact do you expect to see in gut microbiomes, body mass, survivorship with and without infection, oocysts per midgut (or other measures of immune system, and all those things with mushrooms when you expose your models to glyphosate and AMPA? Your Intro should be a review of THAT literature. Since you have the insect apocalypse in your abstract, that should be in your Intro too.

The Introduction needs to be completely revised.

Results:

Given that PLOS Biology aims its papers are a more general readership than other more specialized journals, I recommend that you use fewer acronyms. PO, DQ, GLYPH, MBTH. Just use the real words. The readers don’t have to translate them continually and that improves readership.

I think that experiments 1-4 used some sort of mushroom model and that the remaining experiments use insect models. The results section might benefit from giving some sort of overview as to what experiments were run and what kind of model was used. (The Intro should cover that too, as well as tell us why.)

Much of the paragraph starting on line 105 doesn’t belong in Results. It belongs both in the Introduction and the Materials and Methods. Results sections should include only results.

Line 108, you have a sentence that begins with “First” but there is no second.

Actual results start on line 118. I am not an expert on methods of evaluating melanization. I can clearly see the dose dependent inhibition in Figure 1A, but I cannot understand this experiment given the Results and Figure 1A. A quick word-based sketch of what this experiment was supposed to do and show would benefit your readers. Something was tyrosinase mediated and something else was auto-mediated, but I can’t figure out your model. You should be able to communicate this model to other glyphosate specialists who are unfamiliar with mushrooms and melanization. Also, on line 120, you state that it appears that the inhibition in the reaction is due to inhibited background auto-oxidation. Please explain how you come to that conclusion. Why did you run the experiment with and without tyrosinase?

Section starting line 124: Clarifying as described in the paragraph immediately preceding this paragraph will benefit readers of your second experiment. Why do we care about the rate and level of DC produced? What does it tell us?

The experiment describe starting line 150 is really cool! The authors looked at non-phosphate analogs of glyphosate on DQ (melanin) production and compared them to phosphate-based analogs and glyphosate itself. Why did you home in on P? That should be in your introduction. Make non-experts care about this experiment. Make it clear that measuring DQ is a measure of melanin. Did you still use the mushroom model? Even adding a mushroom icon to your figures might help communicate your model more effectively. In 2F, consider putting the names of the compounds under their structures. Figuring out the name of each compound and finding it on the curves was nearly impossible.

Line 163, you state, “GLYPH and similar compounds inhibit melanin in a non-enzymatic fashion.” If I understand the experiment correctly, you selected phosphate-based compounds and non-phosphate-based compounds to determine which part of the glyphosate molecule drove the effect. (If I didn’t understand the experiment, please clarify it.) In your results, you need to describe the bimodal distribution—state that it’s based on P, which is pretty cool.

Line 214. You need one more sentence that interprets what the result means.

Line 339. Consider leading your results section with the insect models and then moving to the mushroom models.

Line 357: If you’re going to use AMPA in your experiments (and I agree you should!), you need to discuss it in your Introduction.

Line 358. You call out Fig 5 for the first time, after Fig 6. Noooo.

Lines 353-354 and 357-358 are highlights of this paper. They’re the reason I agreed to review this manuscript, and they’re the reason people will want to read the paper. You need to use all of your skills as writers to make your readers understand why those are interesting questions and why it is amazingly cool that you found your answers. Your Intro should be written so as to make me want to know what the impact of glyphosate and AMPA are on insect melanin production.

In Figure 6, you misspelled Fisher.

Lines 419-424. You need to get into the hormesis literature. My apologies. It isn’t fun.

Your experiments regarding insects are easier to understand and interpret than your results with the fungi. You need to set up the fungi experiments better. Something like, once we found these cool results with bugs, we wanted to find the mechanisms so we did this cool thing with a mushroom model. Because of this, please consider leading your results with the insect experiments and leading people to care about the mushroom results. It’ll make for a better read. Leading with the mechanism is slightly painful to read.

How close are the glyphosate doses you used in your insect experiments to those that they may encounter in nature? I mean when you injected the glyphosate.

DISCUSSION:

Line 485: what is it?

You have a paragraph on page 21 that covers almost an entire page. No one can read that. In fact, I recommend that you do through your Discussion and make sure all of your paragraph are no more than 1/3rd of a page. In their current form, your paragraphs are driving away your readers, which will not improve the number of people who reference this awesome paper.

Your opening sentence of that paragraph (line 520) doesn’t make as much sense as one might hope. Your phrase, “which are necessary…” modifies solution but I think you want it to modify free radicals but maybe you want it to modify inhibitors. Line 524, the comma should be a period.

Lines 558-560. I object to using the same suite of references to make the point that glyphosate changes ecosystems by disturbing microbial populations AND inducing oxidative stress. Those should be two sets of references. ALSO, there are many, many papers that show that glyphosate has no impact or only a fleeting impact on ecosystems. (Most of those paper measure microbial biomass rather than community structure, but still, you should acknowledge that.) See for instance: Gornish, E.S., Franklin, K., Rowe, J. and Barberán, A., 2020. Buffelgrass invasion and glyphosate effects on desert soil microbiome communities. Biological Invasions, pp.1-11. Also, read the Discussion of Pochron et al. 2020 for the nuances involved in predicting the soil microbial response to glyphosate and Roundup exposure. The jury is still out on how bad glyphosate is for microbial systems.

Lines 580-581: Body weight does not always respond to exposure to contamination by decreasing (e.g. Pochron et al. 2018; 2019); in fact, sometimes organisms increase their body size in response to exposure to various contaminants, leading to an entire body of literature dedicated to hormesis, defined as a favorable biological response to exposure to toxins or other stressors (Agathokleous et al. 2019; Docea et al. 2019; Wang et al. 2018). Hormesis is most likely to occur at low dose exposures, and it doesn’t involve just glyphosate. The literature includes examples involving heavy metals and parasites. You are not alone in your vexing result regarding body weight.

Line 564: This paragraph is too long.

Line 608: You spend a lot of time talking about concentrations in run off. You’ve missed the literature showing that glyphosate and AMPA occur in our rain, soil, sediments. It would be nice if you could tie that universal contamination in to the ecosystem services delivered by insects. I know that no one will mourn the death of mosquitos, but they are good models for more beneficial insects. You can draw that line. I know nothing about your moth, but if its larva generally lives in the soil, then there is a whole body of literature about the ecosystem services delivered by soil-dwelling insects. People are going to read this paper because of the insect apocalypse angle. You’ve buried that lead in your text. I recommend that you bring it out front and center and make the mechanism the part of your manuscript that gives teeth to your experiments showing that glyphosate is bad for insects. Right now, this is paper is no fun to read. You can make it a lot better by focusing your Introduction and Discussion.

---

## [Decision Letter · Decision Letter 2]

8 Jan 2021

Dear Arturo,

Thank you very much for submitting a revised version of your manuscript "Glyphosate Inhibits Melanization and Increases Insect Susceptibility to Infection" for consideration as a Research Article at PLOS Biology. Thank you also for your patience as we completed our editorial process over the holiday period. This revised version of your manuscript has been evaluated by the PLOS Biology editors, the Academic Editor and the three original reviewers.

In light of the reviews (attached below), we are pleased to offer you the opportunity to address the remaining points from the reviewers in a revised version that we anticipate should not take you very long. We will then assess your revised manuscript and your response to the reviewers' comments and we may consult the reviewers again.

You will see that Reviewer 1 would like you to justify why the experiments with G. mellonella hemolymph were ex-vivo instead of in vivo, or the incubation of mosquito homogenates with glyphosate to analyse melanization inhibition by glyphosate. Reviewer 2 thinks you should tone down your claims given that you have not justified that the conclusions are true for insects in general. Although we will not make it a strict requirement for publication, being able to demonstrate the effects on other insects in vivo would significantly strengthen the work and justify your claims. We would strongly recommend that you try to address this outstanding issue to limit the amount of toning down that would be required.

In addition, we would like to make a suggestion to improve the title changing it to: "Glyphosate inhibits melanization and increases susceptibility to infection in insects"

We expect to receive your revised manuscript within 1 month.

**IMPORTANT - SUBMITTING YOUR REVISION**

3. Resubmission Checklist

a) *Published Peer Review*

b) *PLOS Data Policy*

Please provide the data underlying the following figures, and make sure you mention in the corresponding figure legends where the data can be found:

Fig. 1A-D; Fig. 2A-C; Fig. 3A-D; Fig. 4B-D; Fig. 5A-E; Fig. 6A-D; Fig. 7A-E, H; Fig. S1A-I; Fig. S2A-D; Fig. S3A-C; Fig. S5; Fig. S6A; Fig. S7A, B and Fig. S8A, B

Please don't hesitate to contact us if you have any questions or comments.

Sincerely,

Ines

--

Ines Alvarez-Garcia, PhD,

Senior Editor,

PLOS Biology

Reviewers’ comments

Rev. 1:

In this new submission, Smith and colleagues took into consideration previous comments made by the reviewers and presented a completely reorganized version of the manuscript (which is better in my opinion, and thanks to reviewer #3 comments, which provided insightful comments to restructure the manuscript). They also provided additional data, strengthening some of their findings.

I am satisfied with their responses, and agree with their modifications and most of the statistical analyses provided. I have some additional questions and comments, but they are all minor considering the overall quality of the study.

One thing that called my attention while reading the manuscript for the second time was the performance of an ex vivo, instead of in vivo, experiment with G. mellonella hemolymph, or the incubation of mosquito homogenates with glyphosate to investigate melanization inhibition by glyphosate. I wonder why you chose this methodology, instead of performing a "more realistic" experiment by exposing the larvae/mosquito to glyphosate?

Why performing different statistical tests for similar experiments, such as the survival curve experiments presented in Fig 1C and Fig 2C?

Fig 1D,E: what's the concentration of glyphosate used here? Is it the same as for Fig 1C?

Fig1D FigS1G-I: here you pool data from three distinct experiments that do not show the same trend and do not consider that source of variation in your statistical analyses. I think you should give special attention to this.

The end of lines 153-154 seems to be out of context, since you have only discussed the effects of glyphosate on G. mellonella by that point.

Lines 46 and 244: since you opted to remove most abbreviations, I think it's unnecessary to abbreviate dopaquinone at these two instances, as well as DHI at lines 47 and 328

Line 69: correct the sentence "responsible for amino acid synthesis IS many plants..."

Figure 2C - I could not find information regarding the way the experiment was conducted: were the mosquitoes kept in cages? If so, how many mosquitoes per cage? If there was more than one mosquito per cage and they were all counted for the survival curves, mosquitos in each cage should be considered as pseudoreplicates and this needs to be considered while performing the statistical analysis.

Figure 3D legend says that each treatment group represents 5 individual mosquito midguts, but the pcoa plot shows more than 5 dots for the control and treatment groups.

Line 185: clarify what you mean when you mention "incubated"

Line 210: wrong figure citation: it should be Supplementary Fig. 2C, as in Line 214: wrong figure citation: it should be Supplementary Fig. 2D

Figure 4 - Indicate the concentration of L-DOPA used in the experiments.

Rev. 2:

The authors have reorganized the manuscript so that the insect studies with one new data set are now first, and second is the biochemical characterization of how glyphosate affects the melanization reaction of a fungal tyrosinase. More background is presented for the insect work, and the writing and organization is more comprehensible. Still, as I wrote before, here are two very different types of experimental studies and no substantive basis for their linkage.

Fig. 1-3 present the data for the insect studies, and the effects of glyphosate are largely overly hyped in the Discussion for significance to global insect life and immunity to pathogens. Much ado about little, given that Galleria larvae are highly specialized honey bee pests and not representative of Lepidoptera in general and the overall non-significance of glyphosate treatments on Plasmodium infection and gut microbiota in Anopheles mosquitoes. There are lots more experimental results that appear to be solid for the glyphosate/fungal tyrosinase studies presented in Fig. 4-7. A link would be to show that an isolated insect tyrosinase/phenoloxidase involved in the melanization immune response acts similarly. Sure glyphosate at pharmacological/toxic levels affects "phenoloxidase" activity in unpurified larval hemolymph and ground up mosquitoes as likely would many other widely used chemicals for plant management. It is toxic speculation that "glyphosate inhibits melanization and increases INSECT susceptibility to infection" as stated in the title and all too much so in the Discussion, maybe this can be weakly stated for Galleria larvae but certainly not all INSECTS!

Rev. 3:

Abstract

Line 1: Much improved title.

Line 14: Change to: Melanin, a black-brown pigment found throughout all kingdoms of life, has diverse biological functions including: UV protection, thermoregulation, oxidant scavenging, arthropod immunity, and microbial virulence.

Line 28: Change to: We showed that glyphosate acts as a synergistic antioxidant and disrupts the oxidation-reduction balance of melanization, elucidating the mechanism by which glyphosate inhibits melanization.

Overall, much improved.

Introduction

Line 39: Change to: Melanin, a black-brown pigment found throughout all kingdoms of life, has diverse biological functions including: UV protection, thermoregulation, oxidant scavenging, arthropod immunity, and microbial virulence.

Line 65: Change to: One ubiquitous chemical found in the environment is glyphosate, the most commonly used herbicide world-wide, which was previously shown to interfere with melanization in the fungus Cryptococcus neoformans (26).

Line 89: Put a return before the period to make a new para.

Line 127: This paragraph is not part of an Intro. It’s part of a Results or Conclusion. Remove it or move it.

I’d add chelation to the key words. I’d also add hormesis. You can remove Latin names since they’re in the Abstract.

Overall, much improved!

Results

Line 145: First comma should be a semicolon.

Line 160: Define PBS

Line 227: Missing a cap.

Line 320 and others regarding copper and glyphosate. I recommend citing:

Zhou, C.F., Wang, Y.J., Li, C.C., Sun, R.J., Yu, Y.C. and Zhou, D.M., 2013. Subacute toxicity of copper and glyphosate and their interaction to earthworm (Eisenia fetida). Environmental Pollution, 180, pp.71-77.

Overall, much improved. Your paper is much easier to read and much easier to interpret.

Discussion:

Line 387: Your verb tenses do not agree. Investigated and found—not find.

Line 387:400. Great summary of your work. So much better.

Line 414: This paragraph is really interesting and important. I think I’d play up the public-health angle and see if you can work that point into the abstract or the key words. This is a packed paper, and you might not easily have room, but the fact that glyphosate exposure might make mosquitos more like to transmit plasmodium is really interesting and important.

Line 446: Again, really interesting results. Can you provide a list of the gut microbial species that increased and a list of species that decreased in the supplementary material?

Line 490: Again, you can reference that worm paper regarding copper and glyphosate here.

Summary:

In my opinion, this manuscript could be published with the minor revisions suggested above. The authors made all of the recommended changes in the previous review, and the manuscript is now readable and clear.

---

## [Editor Report · Decision Letter 3]

26 Feb 2021

Dear Arturo,

Thank you for submitting your revised Research Article entitled "Glyphosate Inhibits Melanization and Increases Susceptibility to Infection in Insects" for publication in PLOS Biology. I have now obtained advice from the academic editor and have discussed the revision with the other editors. 

Based on this discussion, we will probably accept this manuscript for publication, provided you satisfactorily address the remaining data requests.

We expect to receive your revised manuscript within two weeks. 

-  a cover letter that should detail your responses to the editorial requests.

*Published Peer Review History*

*Early Version*

Sincerely,

Ines

--

Ines Alvarez-Garcia, PhD,

Senior Editor,

ialvarez-garcia@plos.org,

PLOS Biology

Fig. 1A-D; Fig. 2A-D; Fig. 3A-D; Fig. 4B-D; Fig. 5A-E; Fig. 6A-D; Fig. 7A-E, H; Fig. S1A-I; Fig. S2A-D; Fig. S3A, D, E; Fig. S4; Fig. S5; Fig. S6A; Fig. S7A, B and Fig. S8A, B

---

## [Editor Report · Decision Letter 4]

11 Mar 2021

Dear Dr Casadevall,

On behalf of my colleagues and the Academic Editor, Anita Sil, I am pleased to say that we can in principle offer to publish your Research Article "Glyphosate Inhibits Melanization and Increases Susceptibility to Infection in Insects" in PLOS Biology, provided you address any remaining formatting and reporting issues. These will be detailed in an email that will follow this letter and that you will usually receive within 2-3 business days, during which time no action is required from you. Please note that we will not be able to formally accept your manuscript and schedule it for publication until you have made the required changes.

PRESS

Thank you again for supporting Open Access publishing. We look forward to publishing your paper in PLOS Biology. 

Sincerely, 

Ines

--

Ines Alvarez-Garcia, PhD 

Senior Editor 

PLOS Biology